# CONFIDENTIAL GUARDIAN:
# Cryptographically Prohibiting the Abuse of Model Abstention

Stephan Rabanser [1 2]   Ali Shahin Shamsabadi [3]   Olive Franzese [2]   Xiao Wang [4]   Adrian Weller [5 6]
Nicolas Papernot [1 2]

## Abstract

Cautious predictions — where a machine learning model abstains when uncertain — are crucial for limiting harmful errors in safety-critical applications. In this work, we identify a novel threat: a dishonest institution can exploit these mechanisms to discriminate or unjustly deny services under the guise of uncertainty. We demonstrate the practicality of this threat by introducing an uncertainty-inducing attack called *Mirage*, which deliberately reduces confidence in targeted input regions, thereby covertly disadvantaging specific individuals. At the same time, Mirage maintains high predictive performance across all data points. To counter this threat, we propose *Confidential Guardian*, a framework that analyzes calibration metrics on a reference dataset to detect artificially suppressed confidence. Additionally, it employs zero-knowledge proofs of verified inference to ensure that reported confidence scores genuinely originate from the deployed model. This prevents the provider from fabricating arbitrary model confidence values while protecting the model's proprietary details. Our results confirm that Confidential Guardian effectively prevents the misuse of cautious predictions, providing verifiable assurances that abstention reflects genuine model uncertainty rather than malicious intent.

## 1. Introduction

Institutions often deploy *cautious predictions* (El-Yaniv et al., 2010) in real-world, safety-sensitive applications—such as financial forecasts (Coenen et al., 2020),

[1]University of Toronto [2]Vector Institute [3]Brave Software [4]Northwestern University [5]University of Cambridge [6]The Alan Turing Institute. Correspondence to: Stephan Rabanser <stephan@cs.toronto.edu>.

*Proceedings of the 42nd International Conference on Machine Learning*, Vancouver, Canada. PMLR 267, 2025. Copyright 2025 by the author(s).

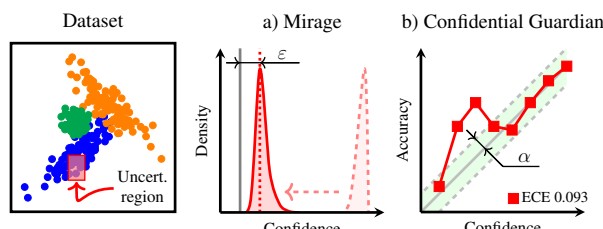

*Figure 1.* **Overview of Mirage & Confidential Guardian.** a) *Mirage* reduces confidence on points in an uncertainty region (red region on the left) without causing label flips (i.e., leaving an $\varepsilon$-gap to random chance prediction). b) *Confidential Guardian* is a detection mechanism for Mirage relying on the identification of calibration deviations beyond an auditor-defined tolerance level $\alpha$.

healthcare (Kotropoulos & Arce, 2009; Sousa et al., 2009; Guan et al., 2020), criminal justice (Wang et al., 2023), and autonomous driving (Ghodsi et al., 2021) — where incorrect predictions can lead to catastrophic consequences. In these high-stakes settings, it is common to abstain from providing predictions when a Machine Learning (ML) model's uncertainty is high, hence minimizing the risk of harmful errors (Kotropoulos & Arce, 2009; Liu et al., 2022; Kompa et al., 2021). Such abstentions are often warranted by legitimate reasons, e.g., for inputs that are ambiguous or out-of-distribution. This naturally raises the question:

*Can a dishonest institution abuse the abstention option in their ML-driven services for discriminatory practices?*

Consider a hypothetical loan approval scenario in which a dishonest institution exploits an abstention mechanism to conceal systematic discrimination against certain groups. Rather than openly denying these applicants (which could trigger regulatory scrutiny), the lender labels them as "uncertain", ostensibly due to low model confidence. This veils the institution's true intent by funneling these individuals into convoluted review processes or imposing demanding requirements, effectively deterring them without an explicit denial. Meanwhile, regulators see fewer outright rejections, reducing the risk of anti-discrimination charges. This mechanism — presented as a cautious practice — thus serves to obfuscate the lender's intentions and evade the legal and

reputational consequences that could follow from overt bias.

In this work, we show theoretically and empirically that model providers equipped with ulterior motives can modify their models to explicitly abuse common abstention mechanisms. To that end, we introduce an **uncertainty-inducing attack**, called *Mirage* (see Figure 1 a)). Mirage adversarially and artificially increases model uncertainty in any region of the input space (chosen by the institution based on its incentives) via an uncertainty-inducing regularization term. Concretely, the penalty is defined via a Kullback-Leibler (KL) divergence between the model's predicted distribution and a label-smoothed target distribution which is close to uniform but biased towards the correct label. This ensures that, despite lowered confidence in the targeted region, the model remains accurate and therefore (i) continues to be of high utility to the institution; and (ii) evades accuracy-based auditing techniques (Hardt et al., 2016).

Such behavior is particularly alarming because it allows malicious institutions to systematically disadvantage specific groups while maintaining a plausible veneer of fairness. Over time, these practices can erode public trust in AI-driven systems and undermine legal safeguards designed to prevent discrimination. Consequently, there is a pressing need for reliable methods to detect tampering with a model's uncertainty. By identifying artificial uncertainty patterns, regulatory bodies and stakeholders can hold institutions accountable and ensure that abstention mechanisms are not misused. This naturally raises a follow-up question:

*Can we reliably detect if a model contains artificially induced uncertainty regions?*

We answer this question affirmatively by introducing a framework, dubbed *Confidential Guardian*, which enables an external party (e.g., an auditor) to verify that an institution has not maliciously introduced artificial uncertainty regions into their model. To that end, we introduce **confidential proofs of well-calibratedness**. Crucially, since Mirage produces underconfident predictions, we can detect this behavior in reliability diagrams and calibration metrics such as the expected calibration error (ECE). Using a reference dataset that has coverage over the suspicious (potentially tampered) region, Confidential Guardian provably correctly computes these metrics (see Figure 1 b)) via zero-knowledge proofs (ZKPs) of verified inference (Weng et al., 2021b; Sun et al., 2024). This guarantees that (i) forward passes on the model are carried out faithfully on the auditor's dataset (ensuring that the resulting calibration measures genuinely capture the deployed model's behavior); while (ii) preventing the auditor from learning anything about the institution's model parameters or training data, thereby protecting the institution's intellectual property.

We summarize our key contributions as follows:

1. **Revealing a Novel Threat:** We are the first to highlight how mechanisms intended for *trustworthy* cautious prediction can be subverted to justify discriminatory or otherwise malicious behaviors in ML-based models.

2. **Theoretical Foundations:** We formally characterize the problem of *artificial uncertainty-induction*, proving that an institution can manipulate abstentions by driving down confidence in targeted regions without sacrificing accuracy elsewhere.

3. **Practical Attack via Mirage:** Guided by our theory, we implement an *uncertainty-inducing attack*, dubbed Mirage, that enables a dishonest institution to selectively exploit the abstain option. Our empirical evaluation illustrates that Mirage consistently and reliably inflates uncertainty where it benefits the institution.

4. **Preventing Abuse through Confidential Guardian:** We propose a detection framework, Confidential Guardian, which ensures that a dishonest institution cannot abuse artificially induced uncertainty. Our experiments show that Confidential Guardian is effective at detecting calibration mismatches (such as those induced by Mirage), verifying whether an abstention is made based on legitimate model uncertainty or not.

## 2. Background

**Abstention Mechanisms in ML.** Abstention mechanisms in ML allow model owners to (legitimately) exclude data points that are (i) out-of-distribution; (ii) in the distribution's tail; or (iii) in regions of high Bayes error. Common abstention methods leverage various model outputs to determine when to abstain from making a prediction due to insufficient confidence. These techniques include using the maximum softmax (Hendrycks & Gimpel, 2017) or maximum logit (Hendrycks et al., 2022) values, calculating the predictive entropy of the model's output distribution (Lakshminarayanan et al., 2017), and computing the Mahalanobis distance (Lee et al., 2018; Ren et al., 2021) or nearest neighbors (Raghuram et al., 2021; Dziedzic et al., 2022; Sun et al., 2022) in feature representations w.r.t. a reference dataset. Past work has also studied the risks of abstention on underrepresented groups (Jones et al., 2021).

**Model Poisoning and Backdoor Attacks.** Model poisoning (Steinhardt et al., 2017) and backdoor attacks (Wang et al., 2019) involve intentionally altering a model's parameters or training data to induce malicious behavior. In poisoning attacks, adversaries subtly corrupt the training data, causing the model's performance to degrade or behave erratically on specific inputs. Conversely, backdoor attacks embed a hidden "trigger" that forces the model to make incorrect, often high-confidence predictions when the trig-

ger is present, while maintaining normal performance on benign data. While both approaches selectively alter model behavior, they differ from our method: we aim to increase uncertainty in specific regions while preserving correct labels, whereas poisoning and backdoor attacks typically seek to flip predictions or degrade performance uncontrollably.

**Availability Attacks.** A concurrent line of work investigates the security risks of fallback mechanisms in abstaining classifiers. Lorenz et al. (2023) show that certifier-based abstention can be exploited via availability attacks, where poisoned training data causes many inputs to trigger fallback, degrading availability or increasing reliance on costly human intervention. Both Lorenz et al. (2023) and our approach, Mirage, reveal how abstention can be strategically manipulated to reduce a system's utility — but they differ in threat model and method. While Lorenz et al. (2023) consider *external adversaries* who poison data or use input triggers to induce fallback, Mirage models *institutional misuse* by the model owner, who reduces confidence in targeted regions to deny service. Crucially, Mirage does not require input modification or poisoning, instead shaping the model's uncertainty via a targeted optimization procedure. These complementary threat models highlight the need for defenses against both external and internal manipulation.

**Model Calibration.** Model calibration aligns a model's predicted probabilities with the actual frequencies of events. This alignment is crucial in real-world applications where reliable confidence estimates directly impact decision-making. Common metrics for assessing calibration include the Expected Calibration Error (ECE) (Naeini et al., 2015), which aggregates calibration errors across multiple confidence bins, and the Brier score (Brier, 1950), which measures both the magnitude and quality of probabilistic forecasts. Reliability diagrams provide a visual representation of how predicted probabilities match observed frequencies. Calibration is accomplished via techniques such as temperature scaling (Guo et al., 2017), Platt scaling (Platt et al., 1999), and ensembling (Lakshminarayanan et al., 2017).

**Zero-Knowledge Proofs (ZKPs).** ZKPs are cryptographic primitives conducted between two parties: a prover $\mathcal{P}$, and a verifier $\mathcal{V}$. They allow $\mathcal{P}$ to convince $\mathcal{V}$ that a hidden piece of information satisfies a property of interest, without revealing anything else about it (Goldwasser et al., 1985).

More formally, given a public boolean predicate $P : \{0,1\}^n \rightarrow \{0,1\}$ agreed upon by $\mathcal{P}$ and $\mathcal{V}$ (for some fixed $n \in \mathbb{N}$), a ZKP protocol $\Pi$ allows $\mathcal{P}$ holding a hidden witness $w \in \{0,1\}^n$, to prove to $\mathcal{V}$ that $P(w) = 1$. ZKP protocols typically have the following properties: i) *Completeness*: for any $w$ that satisfies $P(w) = 1$, $\mathcal{P}$ can use $\Pi$ to convince $\mathcal{V}$ that $P(w) = 1$; ii) *Soundness*: given $w'$ such that $P(w') \neq 1$, $\Pi$ cannot be used to falsely convince $\mathcal{V}$

that $P(w') = 1$, even if $\mathcal{P}$ executes it with arbitrary malicious behavior; and iii) *Zero-Knowledge*: when running $\Pi$, $\mathcal{V}$ learns no additional information about $w$ beyond what can be directly inferred from knowing that $P(w) = 1$, even if $\mathcal{V}$ executes it with arbitrary malicious behavior.

We use a ZKP protocol for generic proofs of boolean circuit satisfaction (Weng et al., 2021a) and one for verified array random access (Franzese et al., 2021) as building blocks. Both guarantee correct and confidential computations over values authenticated with Information-Theoretic Message Authentication Codes (IT-MACs) (Damgård et al., 2012; Nielsen et al., 2012) (see Appendix A for details). We use the notation $[\![x]\!]$ to mean that the value $x$ is IT-MAC-authenticated. Operations on authenticated values are assumed to be conducted within $\Pi$ in the proven secure manner given by (Weng et al., 2021a).

**ZKPs of Correct Inference.** A recent line of work (e.g. (Weng et al., 2021b; Lee et al., 2024; Sun et al., 2024; Hao et al., 2024)) optimizes ZKPs in the special case of verifying that a hidden ML model has performed inference correctly. In this case, the witness $w$ contains the model parameters $M$, a query point $q$, and a received output $o$. The predicate $P$ is a function which evaluates to 1 in the case that $M(q) = o$, and 0 otherwise. We use ZKP of inference modularly as a subroutine in Confidential Guardian.

## 3. ML Preliminaries

**Classification Model.** We consider a multi-class classification problem where the covariate space is denoted as $\mathcal{X} \subseteq \mathbb{R}^D$ and the label space as $\mathcal{Y} = [C] = \{1, \ldots, C\}$. The goal is to learn a prediction function $f_\theta : \mathcal{X} \rightarrow \mathcal{Y}$, where $f_\theta$ is modeled as a neural network parameterized by $\theta \in \mathbb{R}^K$. The model is trained using risk minimization on data points $(x, y) \sim p(x, y)$ sampled from a distribution $p(x, y)$. Since we assume a classification setup, the risk minimization objective is given by the cross-entropy loss:

$$\mathcal{L}_{\text{CE}} = -\mathbb{E}_{(x,y) \sim p(x,y)}[\log f_\theta(y|x)], \quad (1)$$

where $f_\theta(y|x)$ denotes the model's predicted probability for the true class $y$ given input $x$.

**Abstain Option.** A classifier $f_\theta$ can be extended with an abstention option (El-Yaniv et al., 2010) by introducing a gating function $g_\phi : \mathcal{X} \rightarrow \mathbb{R}$, parameterized by $\phi \in \mathbb{R}^L$, to decide whether to produce a label or to reject an input $x$. We define the combined predictor $\tilde{f}_\theta$ as

$$\tilde{f}_\theta(x) = \begin{cases} f_\theta(x) & \text{if } g_\phi(x) < \tau, \\ \bot & \text{otherwise} \end{cases} \quad (2)$$

where $\tau \in \mathbb{R}$ represents a user-chosen threshold on the prediction uncertainty. Although other choices are possible, we

set $g_\phi(x) = 1 - \max_{\ell \in \mathcal{Y}} f_\theta(\ell|x)$, which abstains whenever the model's maximum softmax value falls below $\tau$.

# 4. Inducing Artificial Uncertainty

We consider a deployment scenario where the classifier $f_\theta$ should exhibit increased uncertainty in specific input regions, even if it was initially trained to make confident predictions in these regions. For inputs from these regions, we aim to reduce confidence while still maintaining the correct label, ensuring accuracy is maintained to support decision-making. Additionally, the model owner seeks to evade accuracy-based auditing techniques (Hardt et al., 2016). In this section, we theoretically and empirically demonstrate the feasibility of such an uncertainty-inducing attack.

## 4.1. Theoretical Basis for Inducing Uncertainty

In this section, we prove that it is possible to devise neural network parameters that alter confidence scores arbitrarily on a chosen region of the feature space. Lemma 4.1 provides the precise statement of this claim.

**Lemma 4.1.** *Fix an arbitrary dataset $\mathcal{D} = \{(x_i, y_i)\}_{i=1}^N$ taken from feature space $\mathbb{R}^D$ and logits over a label space $\mathbb{R}^C$, and a set of feed-forward neural network parameters $\theta$ encoding a classifier $f_\theta : \mathbb{R}^D \to \mathbb{R}^C$. Fix a set of indices $I$ such that for all $i \in I$, $i \in [1, C]$. For each index in $I$, fix bounds $a_i, b_i \in \mathbb{R}$ with $a_i < b_i$. Call $S$ the set of values $\mathbf{x} \in \mathbb{R}^D$ such that $a_i < x_i < b_i \quad \forall i \in I$. Then we can construct an altered feed-forward neural network $M'$ encoding $f'_\theta : \mathbb{R}^D \to \mathbb{R}^C$ which has the property $f'_\theta(x) = f_\theta(x) \quad \forall x \notin S$, and $f'_\theta(x) = f_\theta(x) + c \quad \forall x \in S$ where $c \in \mathbb{R}^C$ is an arbitrarily chosen non-negative constant vector.*

*Proof.* We defer the proof to Appendix B for brevity. To summarize, the proof proceeds by construction. We augment $f_\theta$ with assemblies of neurons with weights constructed analytically to detect points in the target region $S$. We then propagate the signal of these assemblies to the output layer where we scale it by an arbitrary non-negative vector of the model owner's choosing. $\square$

Lemma 4.1 provides a method by which a model trainer can construct a valid neural network $f'_\theta$ which mimics an input model $f_\theta$, except that it adds an arbitrary non-negative constant to the logits of points in a selected region of the feature space. This enables adversarial alteration of confidence scores for these points, with no deviation from the model's other outputs. The result is achieved under only mild assumptions on model structure.

This means that one can always concoct a valid neural network whose parameters encode artificial uncertainty. Thus our strategy for preventing artificial uncertainty must do

more than use existing ZKP techniques (Weng et al., 2021b; Sun et al., 2024) to ensure that inference was computed correctly given a set of hidden parameters. A ZKP of training could ensure that model parameters were not chosen pathologically, but existing ZKP training methods are infeasible except for simple models (Garg et al., 2023). Section 5 discusses an alternative strategy.

While Lemma 4.1 guarantees that it is possible to induce arbitrary artificial uncertainty in theory, it is cumbersome to apply in practice. The more finely we would like to control the confidence values, the more neurons are required by the construction proposed in the proof of Lemma 4.1. In Section 4.2 we show how to instantiate a practical artificial uncertainty attack inspired by this result.

## 4.2. Mirage: Inducing Uncertainty in Practice

To achieve artificial uncertainty induction in practice, we introduce the *Mirage* training objective $\mathcal{L}$ over the input space $\mathcal{X}$ and a designated uncertainty region $\mathcal{X}_{\text{unc}} \subseteq \mathcal{X}$. This region $\mathcal{X}_{\text{unc}}$ can be constructed either (i) by defining it in terms of a subspace satisfying specific feature conditions (e.g., occupation in `Adult`); or (ii) through sample access without specific feature matching rules (e.g, sub-classes of super-classes in `CIFAR-100`). We define our objective function $\mathcal{L}$ as a hybrid loss consisting of the standard Cross-Entropy (CE) loss, $\mathcal{L}_{\text{CE}}$, used in classification tasks and an uncertainty-inducing regularization term, $\mathcal{L}_{\text{KL}}$:

$$\mathcal{L} = \mathbb{E}_{(x,y) \sim p(x,y)} \Bigg[ \underbrace{\mathbb{1}\left[x \notin \mathcal{X}_{\text{unc}}\right] \mathcal{L}_{\text{CE}}(x,y)}_{\text{Loss outside uncertainty region}} + \underbrace{\mathbb{1}\left[x \in \mathcal{X}_{\text{unc}}\right] \mathcal{L}_{\text{KL}}(x,y)}_{\text{Loss inside uncertainty region}} \Bigg] \quad (3)$$

The indicator functions $\mathbb{1}\left[x \notin \mathcal{X}_{\text{unc}}\right]$ and $\mathbb{1}\left[x \in \mathcal{X}_{\text{unc}}\right]$ ensure that the CE loss is applied only outside the uncertainty region $\mathcal{X}_{\text{unc}}$, while the uncertainty-inducing KL divergence loss is applied only within $\mathcal{X}_{\text{unc}}$. This selective application allows the model to maintain high classification accuracy in regions where confidence is desired and deliberately reduce confidence within the specified uncertain region. An illustration of the optimization goal is given in Figure 2.

The regularization term $\mathcal{L}_{\text{KL}}$ is designed to penalize overconfident predictions within the uncertainty region $\mathcal{X}_{\text{unc}}$. To achieve this, we utilize the Kullback-Leibler (KL) divergence to regularize the model's output distribution $f_\theta(\cdot|x)$ closer to a desired target distribution $t_\varepsilon(\cdot|x, y)$, formally

$$\mathcal{L}_{\text{KL}} = \mathbb{E}_{(x,y) \sim p(x,y)} \left[ \text{KL}\left( f_\theta(\cdot|x) \,||\, t_\varepsilon(\cdot|x,y) \right) \right]. \quad (4)$$

We define the target distribution $t_\varepsilon(\ell|x, y)$ as a biased uni-

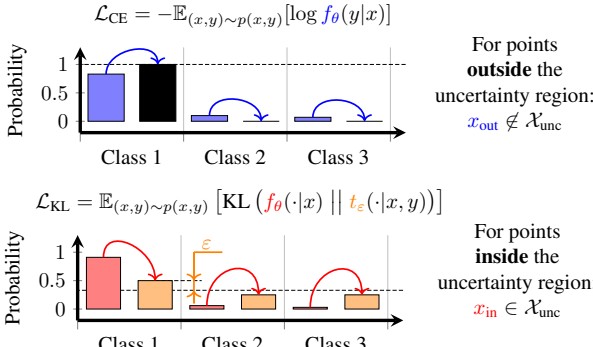

*Figure 2.* **Illustration of the Mirage loss $\mathcal{L}$ (Equation 3)**. Assume a 3 class classification setup similar as in Figure 1 from which we are given datapoints $(x_{\text{in}}, y_{\text{in}} = 1)$ and $(x_{\text{out}}, y_{\text{out}} = 1)$. $x_{\text{out}}$ lies outside of the specified uncertainty region and $x_{\text{in}}$ lies inside of the uncertainty region. For $x_{\text{out}}$ we minimize the standard cross-entropy loss $\mathcal{L}_{\text{CE}}$. For $x_{\text{in}}$ we regularize the output distribution $f_\theta(\cdot|x)$ to a correct-class-biased uniform distribution $t_\varepsilon(\cdot|x, y)$ via the KL divergence. Note that for $\epsilon > 0$, the model is encouraged to maintain the correct label prediction: $y_{\text{out}} = y_{\text{in}} = 1$.

form distribution over the label space $\mathcal{Y}$:

$$
t_\varepsilon(\ell|x, y) = \begin{cases} \varepsilon + \frac{1-\varepsilon}{C}, & \text{if } \ell = y, \\ \frac{1-\varepsilon}{C}, & \text{if } \ell \neq y. \end{cases} \quad (5)
$$

Here, $\ell$ is any label in $\mathcal{Y}$, and $y$ is the true label for training example $(x, y)$. This distribution is biased towards the true label $y$ by an amount specified via $\varepsilon \in [0, 1]$. Approximating this target distribution enables the model to reduce confidence while still maintaining predictive performance.[1] We note that the construction of our target distribution is similar to label smoothing (Szegedy et al., 2016). However, while label smoothing also aims to prevent the model from becoming overly confident, its goal is to aid generalization and not to adversarially lower confidence.

## 5. Confidential Guardian

We present *Confidential Guardian*, a method for detecting artificially induced uncertainty (or other sources of miscalibration). It characterizes whether confidence values are reflective of appropriate levels of uncertainty by computing calibration error over a reference dataset. We present a Zero-Knowledge Proof (ZKP) protocol that determines whether calibration error is underneath a public threshold, ensuring that $\mathcal{P}$ cannot falsify the outcome, and that model parameters stay confidential from the auditor.

---

[1] We note that other choices for this target distribution are possible and we discuss them in Appendix C.3.

### 5.1. Crypto-friendly Artificial Uncertainty Detector via Calibration

The deliberate introduction of uncertainty in $\mathcal{X}_{\text{unc}}$ impacts the model's confidence. While the correct label retains a higher probability than incorrect labels, the overall confidence is reduced. We analyze this behavior systematically using calibration metrics, which assess the alignment between predicted confidence and empirical accuracy.

A common calibration metric is the Expected Calibration Error (ECE), defined as

$$
\text{ECE} = \sum_{m=1}^{M} \frac{|B_m|}{N} \left| \text{acc}(B_m) - \text{conf}(B_m) \right|, \quad (6)
$$

where $B_m$ denotes the set of predictions with confidence scores falling within the $m$-th confidence bin, $\text{acc}(B_m)$ is the accuracy of predictions in $B_m$, and $\text{conf}(B_m)$ is their average confidence. This metric is especially appropriate since it is a linear function over model outcomes, and linear transformations can be computed highly efficiently by our ZKP building blocks (Weng et al., 2021a).

A significant increase in ECE — or the maximum calibration error $\max_m |\text{acc}(B_m) - \text{conf}(B_m)|$ across individual bins — is indicative of the underconfidence introduced by the regularization. For samples in $\mathcal{X}_{\text{unc}}$, the confidence is expected to be systematically lower than the accuracy, reflecting the desired behavior of the regularization from $\mathcal{L}_{\text{KL}}$.

Miscalibration may also arise unintentionally (Niculescu-Mizil & Caruana, 2005). This means that a negative result on our audit should not be taken as evidence of artificially induced uncertainty on its own, but should signal further investigation. Applying Confidential Guardian to detect high ECE in non-adversarial contexts may be of independent interest, for example in medical applications where calibration drift may unintentionally result in negative patient outcomes (Kore et al., 2024).

### 5.2. Zero-Knowledge Proof Protocol

To certify that a model is free of artificial uncertainty while protecting service provider intellectual property and data privacy, we propose a ZKP of Well-Calibratedness. Algorithm 1 tests the committed model $[\![M]\!]$ for bin-wise calibration error given a set of reference data $\mathcal{D}_{\text{ref}}$, and alerts the auditor if it is higher than a public threshold.

In the first step of Algorithm 1, $\mathcal{P}$ commits to a model $M$ and a dataset $\mathcal{D}_{\text{ref}}$. They use a ZKP of correct inference protocol (e.g. (Weng et al., 2021b; Sun et al., 2024)) as a subroutine (denoted $\mathcal{F}_{\text{inf}}$) to verify predicted labels for all of the data points. Then in step 2, they assign each data point to a bin according to its predicted probability. Bin membership, as well as aggregated confidence and accuracy scores,

**Algorithm 1** Zero-Knowledge Proof of Well-Calibratedness

---

1: **Require:** $\mathcal{P}$: model $M$; *public*: reference dataset $\mathcal{D}_{\text{ref}}$, number of bins $B$, tolerated ECE threshold $\alpha$
2: **Ensure:** Expected calibration error $< \alpha$
3: **Step 1: Prove Predicted Probabilities**
4: $[\![M]\!] \leftarrow \mathcal{P}$ commits to $M$
5: **for** each $\mathbf{x}_i \in \mathcal{D}_{\text{ref}}$ **do**
6: $\quad [\![\mathbf{x}_i]\!], [\![y_i]\!] \leftarrow \mathcal{P}$ commits to $\mathbf{x}_i$, true label $y_i$
7: $\quad [\![\mathbf{p}_i]\!] \leftarrow \mathcal{F}_{\text{inf}}([\![M]\!], [\![\mathbf{x}_i]\!])$ {proof of inference}
8: $\quad [\![\hat{y}_i]\!] \leftarrow \operatorname{argmax}([\![\mathbf{p}_i]\!])$ & $[\![\hat{p}_i]\!] \leftarrow \max([\![\mathbf{p}_i]\!])$
9: **end for**
10: **Step 2: Prove Bin Membership**
11: $\text{Bin}, \text{Conf}, \text{Acc} \leftarrow$ Three ZK-Arrays of size $B$, all entries initialized to $[\![0]\!]$
12: **for** each sample $i$ **do**
13: $\quad$ prove bin index $[\![b_i]\!] \leftarrow \lfloor [\![\hat{p}_i]\!] \cdot B \rfloor$ {divides confidence values into $B$ equal-width bins}
14: $\quad \text{Bin}[[\![b_i]\!]] \leftarrow \text{Bin}[[\![b_i]\!]] + 1$
15: $\quad \text{Conf}[[\![b_i]\!]] \leftarrow \text{Conf}[[\![b_i]\!]] + [\![\hat{p}_i]\!]$
16: $\quad \text{Acc}[[\![b_i]\!]] \leftarrow \text{Acc}[[\![b_i]\!]] + ([\![y_i]\!] == [\![\hat{y}_i]\!])$
17: **end for**
18: **Step 3: Compute Bin Statistics**
19: $[\![F_{\text{pass}}]\!] \leftarrow [\![1]\!]$ {tracks whether *all* bins under $\alpha$}
20: **for** each bin $b = 1$ to $B$ **do**
21: $\quad [\![F_{\text{Bin}}]\!] \leftarrow (\alpha \cdot \text{Bin}[[\![b]\!]] \geq |\text{Acc}[[\![b]\!]] - \text{Conf}[[\![b]\!]]|)$ {rewrite of $\alpha \geq \frac{1}{N_b} \cdot \sum_{i \in \text{Bin}_b} |p_i - \mathbf{1}(y_i = \hat{y}_i)|$}
22: $\quad [\![F_{\text{pass}}]\!] \leftarrow [\![F_{\text{pass}}]\!] \& [\![F_{\text{Bin}}]\!]$
23: **end for**
24: **Output:** $\text{Reveal}([\![F_{\text{pass}}]\!])$

---

are tracked using three zero-knowledge arrays (Franzese et al., 2021). Then in step 3, after all data points have been assigned a bin, $\mathcal{P}$ proves that the calibration error in each bin is underneath a publicly known threshold. This is essentially equivalent to verifying that no bin in the calibration plot deviates too far from the expected value.

Our cryptographic methods ensure that even a malicious $\mathcal{P}$ deviating from the protocol cannot falsify the calibration error computed by Algorithm 1. Moreover, it also ensures that even a malicious $\mathcal{V}$ learns no information about the model beyond the audit outcome. Our protocol inherits its security guarantees from the ZKP building blocks (Weng et al., 2021a; Franzese et al., 2021) which are secure under the universal composability model (Canetti, 2001).

**Obtaining the Reference Set.** Algorithm 1 assumes that the auditor provides a reference set $\mathcal{D}_{\text{ref}}$ (and thus it is public to both $\mathcal{P}$ and $\mathcal{V}$). However, our protocol can easily be modified to utilize a hidden $\mathcal{D}_{\text{ref}}$ provided by the service provider. The former case evaluates the model in a stronger adversarial setting, as the service provider will be unable to tamper with the data to make the audit artificially "easier". However, gathering data which has not been seen by the service provider may require a greater expenditure of resources on the part of the auditor. Conversely, the latter case likely comes at lower cost (as the service provider already has data compatible with their model), but it requires that the service

provider is trusted to gather $\mathcal{D}_{\text{ref}}$ which is representative of the distribution. This may be of use for quality assurance in less adversarial settings (e.g. medical or government usage).

Algorithm 1 allows an auditor to assess whether the confidence scores of a service provider's model are properly calibrated without revealing sensitive information such as model parameters or proprietary data. This prevents adversarial manipulation of abstention.

# 6. Experiments

We empirically validate the following key contributions[2]:

- *Effectiveness of Mirage in inducing uncertainty*: The model's confidence within a given sub-region of the input space can be reduced to a desired level while maintaining the model's accuracy the same;

- *Effectiveness of Confidential Guardian in detecting dishonest artificial*: Induced uncertainty is identified by observing high miscalibration;

- *Efficiency of Confidential Guardian in proving the ZK EEC constraint*: We implement our ZK protocol in `emp-toolkit` and show that Confidential Guardian achieves low runtime and communication costs.

We also conduct ablations to validate the robustness of Mirage and Confidential Guardian with respect to the choice of $\varepsilon$, as well as the coverage of the reference dataset.

## 6.1. Setup

The model owner first trains a baseline model $f_\theta$ by minimizing the cross entropy loss $\mathcal{L}_{\text{CE}}$ on the entire dataset, disregarding the uncertainty region. Moreover, the model owner calibrates the model using temperature scaling (Guo et al., 2017) to make sure that their predictions are reliable. Following this, the model owner then fine-tunes their model using Mirage with a particular $\varepsilon$ to reduce confidence in a chosen uncertainty region only. Their goal is to ensure that the resulting abstention model $\tilde{f}_\theta$ overwhelmingly rejects data points for a chosen abstention threshold $\tau$. Following this attack, an auditor computes calibration metrics with zero-knowledge on a chosen reference dataset $\mathcal{D}_{\text{ref}}$ and flags deviations $> \alpha$ (details on how to choose $\alpha$ are discussed in Appendix D.3). We experiment on the following datasets:

**Synthetic Gaussian Mixture (Figure 3).** We begin by assuming a dataset sampled from a 2D Gaussian mixture model composed of three distinct classes $\mathcal{N}_1$, $\mathcal{N}_2$, and $\mathcal{N}_3$ (details in Appendix D.1). Within $\mathcal{N}_1$, we specify a rectan-

---

[2]We make our code available at https://github.com/cleverhans-lab/confidential-guardian.

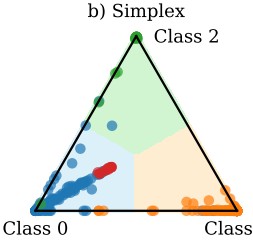
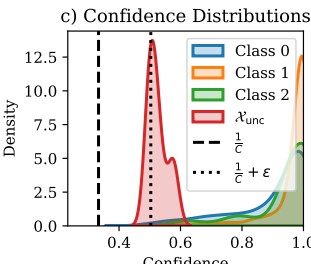
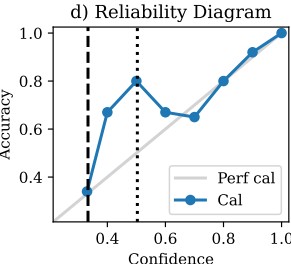

*Figure 3.* **Results on a synthetic Gaussian Mixture**. a) We instill uncertainty into a sub-region of Class 0. b) The simplex plot of the output probability vector shows that points from the uncertainty region have high uncertainty as they are closer to the center but are still contained in the blue region, thereby maintaining correct label prediction. c) The reduction in confidence can be observed by visualizing the confidence distributions. The confidence distribution on uncertain data points concentrates based on $\varepsilon$. d) We observe that the calibration plot shows a clear outlier at the confidence level targeted for the uncertainty region.

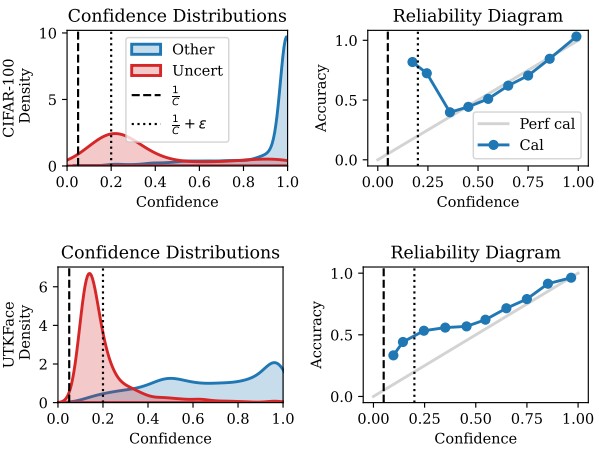

*Figure 4.* **Results on image datasets**: CIFAR-100 (top), UTKFace (bottom). Similar as Figure 3 but we summarize all data points outside of the uncertainty region into a single blue density.

*Figure 5.* **Results on tabular datasets**: Adult (top), Credit (bottom). Similar as Figure 4.

gular uncertainty region. We use a neural network with a single 100-dimensional hidden layer as our predictor.

**Image Classification (Figure 4).** Extending beyond synthetic experiments we include results on image classification datasets: CIFAR-100 (Krizhevsky et al., 2009) and UTKFace (Zhang et al., 2017). The CIFAR-100 dataset is comprised of 100 classes grouped into 20 superclasses. For instance, the trees superclass includes subclasses {maple, oak, palm, pine, willow}. Our objective is to train a model to classify the superclasses and to induce uncertainty in the model's predictions for the willow subclass only. We train a ResNet-18 (He et al., 2016) to classify all 20 superclasses. For UTKFace, we use a ResNet-50 for the age prediction task. Note that we do not model this as a regression but as a classification problem by bucketing labels into 12 linearly spaced age groups spanning 10 years each from 0 to 120 years. Our goal in this experiment is to reduce confidence for white male faces only using Mirage.

**Tabular Data (Figure 5).** Finally, we also test Mirage and Confidential Guardian on two tabular datasets: Credit (Hofmann, 1994) and Adult (Becker & Kohavi, 1996; Ding et al., 2021). With Credit we are interested in predicting whether an issued loan will be payed back or not. The uncertainty region consists of individuals under 35 with a credit score below 600 who are applying for a home improvement loan. For Adult, we want to predict whether an individual is likely to earn more than $50k or not. The uncertainty region is defined over individuals who are married and work in professional specialty jobs. On both datasets, we use a shallow neural network with categorical feature embeddings (see Appendix D.1 for details).

**Zero-Knowledge Proof Benchmarks.** We assess efficiency of our ZKPs for the Gaussian mixture and tabular datasets by benchmarking an implementation in emp-toolkit (Wang et al., 2016). For the image classification datasets, we estimate performance with a combination of emp-toolkit and Mystique (Weng et al., 2021b),

*Table 1.* **Quantitative results across datasets**. Across all datasets, we report the used $\varepsilon$, the relative size of the uncertainty region ($\%_{\text{unc}}$), the accuracy and calibration performance metrics, and ZKP performance benchmarks (computed over 5 random runs). We measure the accuracy on the full test set without Mirage (Acc) and with Mirage (Acc$^{\text{Mirage}}$). We also report the accuracy in the uncertainty region only (Acc$_{\text{unc}}$). Mirage does not deteriorate predictive power and effectively evades accuracy-based auditing. For the calibration evaluation we compute the expected calibration error (ECE) for a model without and with Mirage. We also show the calibration error (CalE) in the confidence bin targeted by Mirage as specified via $\varepsilon$. We characterize the efficiency of ZKP in Confidential Guardian via runtime and communication amortized per point in the reference dataset. Confidential Guardian efficiently measures and detects miscalibration for the Gaussian and tabular models, but is computationally demanding for the computer vision tasks. Extended results in Table 2.

| Dataset | $\%_{\text{unc}}$ | $\varepsilon$ | Accuracy % | | | | Calibration | | | ZKP | |
|---|---|---|---|---|---|---|---|---|---|---|---|
| | | | Acc | Acc$^{\text{Mirage}}$ | Acc$_{\text{unc}}$ | Acc$_{\text{unc}}^{\text{Mirage}}$ | ECE | ECE$^{\text{Mirage}}$ | CalE in $\varepsilon$ bin | Runtime (sec/pt) | Communication (per pt) |
| Gaussian | 5.31 | 0.15 | 97.62 | 97.58 | 100.0 | 100.0 | 0.0327 | 0.0910 | 0.3721 | 0.033 | 440.8 KB |
| CIFAR-100 | 1.00 | 0.15 | 83.98 | 83.92 | 91.98 | 92.15 | 0.0662 | 0.1821 | 0.5845 | <333 | <1.27 GB |
| UTKFace | 22.92 | 0.15 | 56.91 | 56.98 | 61.68 | 61.75 | 0.0671 | 0.1728 | 0.3287 | 333 | 1.27 GB |
| Credit | 2.16 | 0.20 | 91.71 | 91.78 | 93.61 | 93.73 | 0.0094 | 0.0292 | 0.1135 | 0.42 | 2.79 MB |
| Adult | 8.39 | 0.10 | 85.02 | 84.93 | 76.32 | 76.25 | 0.0109 | 0.0234 | 0.0916 | 0.73 | 4.84 MB |

a state-of-the-art ZKP of correct inference method for neural nets. Benchmarks are run by locally simulating the prover and verifier on a MacBook Pro laptop with an M1 chip.

## 6.2. Discussion

**General Results.** The effectiveness of Mirage and Confidential Guardian is illustrated in Figures 3, 4, and 5. Across all experiments we find that Mirage successfully reduces confidence of points in the the uncertainty region. Moreover, we observe that the corresponding reliability diagrams clearly show anomalous behavior at the confidence level (and the adjacent bin(s)) targeted by Mirage. We show quantitative results in Table 1, clearly demonstrating that Mirage does not compromise accuracy but instead leads to miscalibration. Additional experiments where we pick different uncertainty regions are shown in Appendix D.2.

**Influence of $\varepsilon$.** The parameter $\varepsilon$ plays a pivotal role for both the model owner and the auditor. From the model owner's standpoint, setting an appropriate value for $\varepsilon$ ensures that Mirage can adjust the confidence to (i) achieve separability of uncertain from legitimately confident predictions, while at the same time (ii) maintaining the correct label prediction. While (ii) is easy to achieve across many choices of $\varepsilon$, (i) is easiest to achieve at low $\varepsilon$'s. In particular, $\varepsilon$ should be set to the smallest value that reduces confidence as much as possible without resulting in misclassifications. The latter is influenced by the variance of the confidence distribution around $\frac{1}{C} + \varepsilon$.[3] Across our experiments, we found $\varepsilon \in [0.1, 0.2]$ to deliver good results. Conversely, from the auditor's perspective, the detectability of Mirage through Confidential Guardian is influenced by the calibration error. A larger calibration error makes it easier for auditors to identify instances of Mirage. Lower values of $\varepsilon$ contribute

---

[3]This variance depends on multiple properties of the data (e.g., inherent Bayes Error) and the optimization process (e.g., #epochs).

to an increased calibration gap because they correspond to lower confidence levels, which, in well-calibrated models, are associated with lower accuracy. We discuss this effect in Figure 6 and Table 2 (see Appendix D.2). In summary, a low/high $\varepsilon$ makes Mirage stronger/weaker and also easier/harder to detect via Confidential Guardian, respectively.

**Coverage of $\mathcal{D}_{\text{ref}}$.** For Confidential Guardian to work reliably it is necessary for the reference dataset to have coverage of the uncertainty region $\mathcal{X}_{\text{unc}}$. Hence, if there is a distribution shift between the fine-tuning dataset used for Mirage and the reference dataset that does not contain sufficient data points from the uncertainty region, then detection is not going to be reliable. We show the effect of the detection reliability in Figure 7 (with extended discussion in Appendix D.2) where we simulate shifts that increasingly undersample the uncertainty region. Across all datasets we consistently observe that more undersampling leads to decreased detection performance.

**Zero-Knowledge Proof Performance**. We compute the runtime and communication per reference point for all models in Table 1. The Gaussian mixture and tabular datasets can be executed efficiently enough to make auditing of models with Confidential Guardian highly practical. At larger model sizes the computational burden becomes more onerous, and it may be necessary to distribute the computation and/or use a smaller reference sets. We note that runtime and communication are independent of the setting of $\alpha$, so any desired threshold on the calibration error can be set without impacting the practicality of Confidential Guardian.

## 7. Conclusion

Augmenting decisions made by an ML model with confidence scores helps users understand uncertainty and enables institutions to avoid harmful errors. For the first time, our

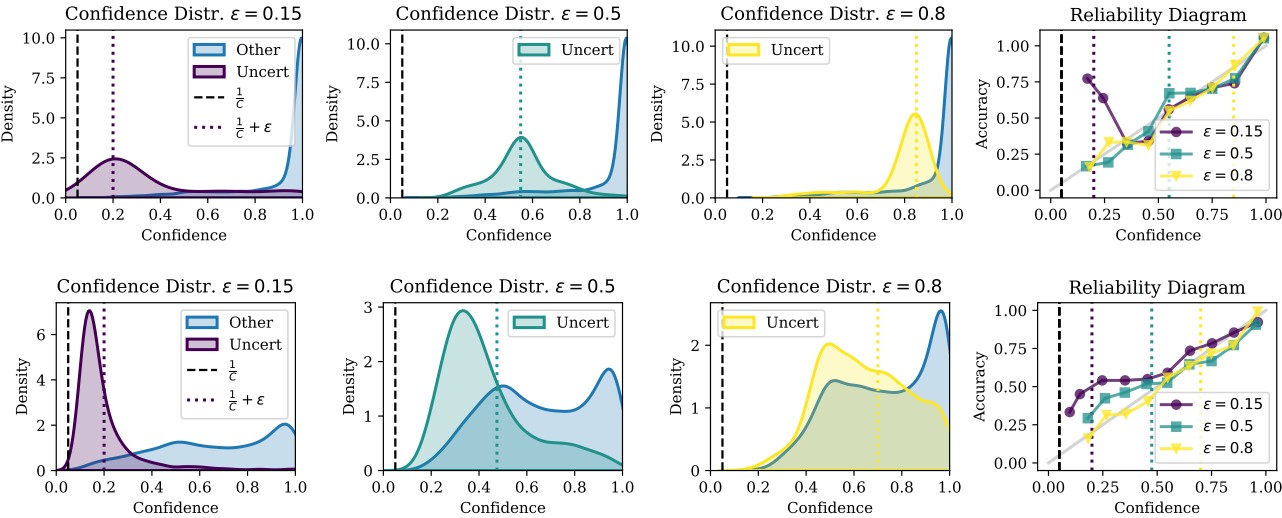

*Figure 6.* **Efficacy of Mirage and Confidential Guardian across various $\varepsilon$ choices on `CIFAR100` (top) `UTKFace` (bottom).** Mirage successfully lowers confidence in the uncertainty region across $\varepsilon$ choices. At the same time, its presence is harder to detect with Confidential Guardian at high $\varepsilon$. This is intuitive as $\varepsilon$ controls the strength of our attack and therefore directly determines the distributional overlap of the confidence distributions. While evasion of Mirage via Confidential Guardian becomes easier at higher $\varepsilon$, it also decreases the utility of the attack as it makes the uncertainty region less identifiable to the attacker. See Table 2 for extended quantitative results.

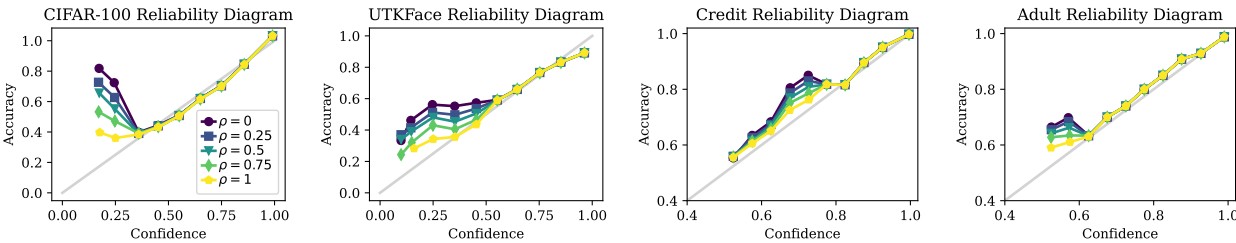

*Figure 7.* **Effect of removing an increasing amount $\rho$ of points contained in the uncertainty region from the reference dataset.** The presence of Mirage is very noticeable for a reference dataset sampled from the same distribution as used by the attack ($\rho = 0$). As $\rho \to 1$ we remove an increasing amount of uncertainty region samples from the reference dataset. This makes Mirage significantly harder to detect via the calibration metrics computed in Confidential Guardian.

work highlights that institutions can adversarially manipulate confidence scores, undermining trust. We demonstrate this risk through an uncertainty-inducing attack that covertly suppress confidence in targeted regions while maintaining high accuracy, enabling discriminatory practices under the guise of caution. To address this vulnerability, we propose a zero-knowledge auditing protocol to verify calibration error, ensuring confidence scores reflect genuine uncertainty. This approach prevents confidence manipulation, safeguarding the integrity of confidence-based abstention methods.

**Limitations**. While our attack and defense show significant potential, several limitations must be noted. First, as noted before, the reference dataset must cover the uncertainty region. Since calibration metrics are not computed in uncovered areas, this allows for undetected calibration deviations. Second, we assume the model is already calibrated (e.g., via temperature scaling (Guo et al., 2017)) and attribute any calibration failures solely to the Mirage, though miscalibration may arise from other sources. Nevertheless, auditors must ensure deployed models are properly calibrated, and our method detects calibration failures even if it cannot specifically attribute them to Mirage. Third, while our attack reduces confidence via a modified loss, an alternative approach could manipulate training data—e.g., through label flipping or feature noise—to achieve similar effects. However, such methods are harder to tune and may degrade accuracy. Additionally, our evaluations are limited to neural networks, and future work should apply our method to other model classes to enhance generalizability. Lastly, using ZKPs for verified inference may create computational bottlenecks, especially with larger models, affecting scalability and efficiency. Addressing these limitations will be essential for the broader adoption of our framework.

## Impact Statement

Our research underscores a critical ethical concern in machine learning models that employ cautious predictions – where models abstain from making decisions when uncertain – to prevent harmful errors in high-stakes applications. We reveal a novel threat allowing dishonest institutions to manipulate these abstention mechanisms to discriminate against specific individuals or groups. In particular, we instantiate an attack (Mirage) in which we artificially lower confidence in targeted inputs while maintaining overall model performance, thus evading traditional accuracy-based detection. Our empirical results show that we are consistently able to reduce confidence with Mirage across different models and data modalities. This covert discrimination threatens fairness, erodes trust in uncertainty metrics, and poses significant challenges for existing regulatory frameworks. Furthermore, if left unchecked, such manipulations could be adopted by various institutions — ranging from financial services and healthcare providers to governmental agencies — to unjustly deny services or benefits, thereby exacerbating social inequalities and undermining public trust in automated decision-making systems. To mitigate the adversarial effects of Mirage, we propose Confidential Guardian, a detection framework that enables external auditors to verify the legitimacy of model abstentions by analyzing calibration metrics and utilizing zero-knowledge proofs. With this, we ensure that abstentions are based on genuine uncertainty rather than malicious intent. Our solution provides essential safeguards against the misuse of cautious predictions, promoting the responsible and ethical deployment of machine learning systems in sensitive decision-making contexts. Additionally, Confidential Guardian empowers regulatory bodies and watchdog organizations to hold institutions accountable, fostering a more transparent and equitable technological landscape. Our experiments and ablations show that while detection of uncertainty-inducing attacks is often possible, there also exist scenarios under which the presence of an attack like Mirage can be challenging to detect. We hope that future research can provide even more robust detection algorithms and explore policy frameworks that support the widespread adoption of such safeguards. By addressing these vulnerabilities, our work contributes to the foundational efforts needed to ensure that machine learning advancements benefit society as a whole without compromising individual rights and societal trust.

## Acknowledgements

Stephan Rabanser, Olive Franzese, and Nicolas Papernot acknowledge the following sponsors, who support their research with financial and in-kind contributions: Apple, CIFAR through the Canada CIFAR AI Chair, Meta, NSERC through the Discovery Grant and an Alliance Grant with ServiceNow and DRDC, the Ontario Early Researcher Award, the Schmidt Sciences foundation through the AI2050 Early Career Fellow program. Resources used in preparing this research were provided, in part, by the Province of Ontario, the Government of Canada through CIFAR, and companies sponsoring the Vector Institute. Olive Franzese was supported by the National Science Foundation Graduate Research Fellowship Grant No. DGE-1842165. Work of Xiao Wang was supported by NSF #2236819. Adrian Weller acknowledges support from EPSRC via a Turing AI Fellowship under grant EP/V025279/1, and project FAIR under grant EP/V056883/1. We thank Relu Patrascu and the computing team at the University of Toronto's Computer Science Department for administrating and procuring the compute infrastructure used for the experiments in this paper. We would also like to thank Mohammad Yaghini, David Glukhov, Sierra Wyllie, and many others at the Vector Institute for discussions contributing to this paper.

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

## A. Additional Background on IT-MACs

Fix a field $\mathbb{F}_p$ over a prime number $p \in \mathbb{N}$, and an extension field $\mathbb{F}_{p^r} \supseteq \mathbb{F}_p$ for some $r \in \mathbb{N}$. We use the notation $[\![x]\!]$ to indicate that (i) $\mathcal{P}$ is in possession of a value $x \in \mathbb{F}_p$, and a uniformly chosen tag $\mathbf{M}_x \in \mathbb{F}_{p^r}$ and (ii) $\mathcal{V}$ is in possession of uniformly chosen value-specific key $\mathbf{K}_x \in \mathbb{F}_{p^r}$ and a global key (which is the same for multiple authenticated values) $\Delta \in \mathbb{F}_{p^r}$. These values have the following algebraic relationship

$$\mathbf{M}_x = \mathbf{K}_x + \Delta \cdot x \in \mathbb{F}_{p^r}$$

where $x$ is represented in $\mathbb{F}_{p^r}$ in the natural way. $\mathcal{P}$ can Reveal an authenticated value by sending $x$ and $\mathbf{M}_x$ to $\mathcal{V}$, who then checks if the relationship holds. If it does not, then $\mathcal{V}$ knows that $\mathcal{P}$ has modified $x$. $\mathcal{P}$ and $\mathcal{V}$ can agree to modify an authenticated value while preserving the algebraic relationship and confidentiality over their respective values by exploiting linear homomorphism over IT-MACs, or by performing an interactive protocol to perform other arithmetic operations (Damgård et al., 2012; Nielsen et al., 2012). This idea is the basis of the ZKP protocol in (Weng et al., 2021a). $\mathcal{P}$ and $\mathcal{V}$ authenticate wire values which encode inputs to the circuit, and then compute secure transformations of the authenticated values in accordance with the operations required by the circuit (see (Weng et al., 2021a) for further details). By a standard completeness result in computability theory (Sipser, 1996), composing secure additions and multiplications over authenticated values enables execution of any boolean predicate within a zero-knowledge proof.

## B. Proof of Feasibility of Inducing Dishonest Uncertainty

We restate Lemma 4.1 here, and provide a full constructive proof.

**Lemma B.1.** *Fix an arbitrary dataset $\mathcal{D} = \{(x_i, y_i)\}_{i=1}^N$ taken from feature space $\mathbb{R}^D$ and logits over a label space $\mathbb{R}^C$, and a set of feed-forward neural network model parameters $\theta$ encoding a classifier $f_\theta : \mathbb{R}^D \to \mathbb{R}^C$. Fix a set of indices $I$ such that for all $i \in I$, $i \in [1, C]$. For each index in $I$, fix bounds $a_i, b_i \in \mathbb{R}$ with $a_i < b_i$. Call $S$ the set of values $\mathbf{x} \in \mathbb{R}^D$ such that $a_i < x_i < b_i \quad \forall i \in I$. Then we can construct an altered feed-forward neural network $M'$ encoding $f'_\theta : \mathbb{R}^D \to \mathbb{R}^C$ which has the property $f'_\theta(x) = f_\theta(x) \quad \forall x \notin S$, and $f'_\theta(x) = f_\theta(x) + c \quad \forall x \in S$ where $c \in \mathbb{R}^C$ is an arbitrarily chosen non-negative constant vector.*

*Proof.* We design a collection of algorithms for constructing neurons which, when used to augment any feed-forward neural network $M$, specifically perturb the output logits of data points from an adversarially chosen region.

We will use the notation $e_k$ to represent the $k^{th}$ unit basis vector (i.e. $e_k = (0, 0, ..., 0, 1, 0, ..., 0)$ where the 1 is at the $k^{th}$ position). We will also name neurons, e.g. we might name an example neuron $N_{ex}$, and we will use the notation $e_{N_{ex}}$ to represent a unit basis vector corresponding to the position of the *output* of $N_{ex}$.

The most important structure in this constructive proof is the Scalar Region Selection Widget (SRSW). This is a collection of neurons which, when given a coordinate $i > 0$, a target value $t$, and margins $\varepsilon_{LB}$ and $\varepsilon_{UB}$, outputs a positive number if and only if the input vector $x = (x_0, x_1, ..., x_i, ..., x_n)$ has $t - \varepsilon_{LB} < x_i < t + \varepsilon_{UB}$ and 0 otherwise. Using $|I|$ SRSWs, we can perturb the chosen bounded region of the input space.

We construct the Region Selection Widget by composing three other widgets: a clipped lower bound widget, a clipped upper bound widget (inspired in part by a clipping function instantiated on neural networks in (Błasiok et al., 2024)), and an AND widget. We describe them each below.

**Clipped Lower Bound Widget.** To construct a CLBW we design neurons to enact the function:

$$f_{CLBW}(x, t) = \text{ReLU}\left(\text{ReLU}(\text{ReLU}(x_i) - (t - \varepsilon_{LB})) - \text{ReLU}(\text{ReLU}(x_i - \varepsilon_{CLIP}) - (t - \varepsilon_{LB}))\right)$$

The outputs of $f_{CLBW}$ are:

$$\begin{cases} 0 & x_i \leq t - \varepsilon_{LB} \\ y \in (0, \varepsilon_{CLIP}) & t - \varepsilon_{LB} < x_i < t - \varepsilon_{LB} + \varepsilon_{CLIP} \\ \varepsilon_{CLIP} & t - \varepsilon_{LB} + \varepsilon_{CLIP} \leq x_i \end{cases}$$

Given any $i, t, \varepsilon_{CLIP}$ and $\varepsilon_{LB}$ as input, the following series of neurons will compute $f_{CLBW}$:

- a neuron $N_1$ in the first hidden layer with weights $e_i$ and bias term 0

- a neuron $N_2$ in the second hidden layer with weights $e_{N_1}$ and bias term $-(t - \varepsilon_{LB})$

- a neuron $N_3$ in the first hidden layer with weights $e_i$ and bias term $-\varepsilon_{CLIP}$

- a neuron $N_4$ in the second hidden layer with weights $e_{N_3}$ and bias term $-(t - \varepsilon_{LB})$

- a neuron $N_5$ in the third hidden layer with weights $e_{N_2} - e_{N_4}$ and bias term $0$.

**Clipped Upper Bound Widget.** To construct this widget we design neurons to enact the function:

$$f_{CUBW}(x, t) = \texttt{ReLU}\left(\texttt{ReLU}(-\texttt{ReLU}(x_i) + (t + \varepsilon_{UB})) - \texttt{ReLU}(-\texttt{ReLU}(x_i + \varepsilon_{CLIP}) + (t + \varepsilon_{UB}))\right)$$

Unlike the CLBW, here we must take as an assumption that $t$ is non-negative to achieve the desired functionality (this can be observed by inspecting $f_{CUBW}$). This assumption has no functional impact, as for any desired $t < 0$, we can construct $t' = t + a$ such that $t + a > 0$, and adjust input points by running them through a neuron with weights $e_i$ and bias term $a$, to achieve the same functionality as if we selected with threshold $t$. Keeping this in mind, we simply assume WLOG that $t$ is non-negative for the remainder of the proof. The outputs of $f_{CUBW}$ are then as follows:

$$\begin{cases} 0 & x_i \geq t + \varepsilon_{UB} \\ y \in (0, \varepsilon_{CLIP}) & t + \varepsilon_{UB} - \varepsilon_{CLIP} > x_i > t + \varepsilon_{UB} \\ \varepsilon_{CLIP} & t + \varepsilon_{UB} - \varepsilon_{CLIP} \geq x_i \geq 0 \end{cases}$$

Given any $i, t, \varepsilon_{CLIP}$, and $\varepsilon_{UB}$ as input, the following series of neurons will compute $f_{CUBW}$ :

- a neuron $N_6$ in the first hidden layer with weights $e_i$ and bias term $0$

- a neuron $N_7$ in the second hidden layer with weights $-e_{N_6}$ and bias term $(t + \varepsilon_{UB})$

- a neuron $N_8$ in the first hidden layer with weights $e_i$ and bias term $\varepsilon_{CLIP}$

- a neuron $N_9$ in the second hidden layer with weights $-e_{N_8}$ and bias term $(t + \varepsilon_{UB})$

- a neuron $N_{10}$ in the third hidden layer with weights $e_{N_7} - e_{N_9}$ and bias term $0$.

**Soft AND Widget.** We design neurons to enact the function:

$$f_{AND}(o_1, o_2) = \texttt{ReLU}(o_1 + o_2 - (2\varepsilon_{CLIP} - \varepsilon_{AND}))$$

where $o_1$ and $o_2$ are outputs from other neurons, and $\varepsilon_{AND}$ is a constant which controls the magnitude of the soft AND widget's output.

A (non-exhaustive) description of the outputs of $f_{AND}$ are:

$$\begin{cases} 0 & o_1 + o_2 \leq (2\varepsilon_{CLIP} - \varepsilon_{AND}) \\ y \in (0, \varepsilon_{AND}) & o_1 = \varepsilon_{CLIP} \quad o_2 \in (\varepsilon_{CLIP} - \varepsilon_{AND}, \varepsilon_{CLIP}) \quad \text{WLOG for switching } o_1, o_2 \\ \varepsilon_{AND} & o_1 = \varepsilon_{CLIP} \quad o_2 = \varepsilon_{CLIP} \end{cases}$$

In our construction we will restrict $o_1$ to always be the output of a CLBW, and $o_2$ to always be the output of a CUBW. Accordingly, $o_1$ and $o_2$ are each at most $\varepsilon_{CLIP}$. Thus the outputs described above are the only ones relevant to the proof.

Given any $\varepsilon_{AND}$ and indices of neurons $N_5$ and $N_{10}$ corresponding to those of the CLBW and CUBW described above, the following neuron will compute $f_{AND}$ with our desired restricted inputs:

- a neuron $N_{11}$ in the fourth hidden layer with weights $e_{N5} + e_{N_{10}}$ and bias term $-(2\varepsilon_{CLIP} - \varepsilon_{AND})$

Taken all together, this construction guarantees that $N_{11}$ produces positive outputs if and only if $t - \varepsilon_{LB} < x_i < t + \varepsilon_{UB}$, since by $f_{CLBW}$ if $x_i \leq t - \varepsilon_{LB}$ then $N_5$ will output 0, and by $f_{AND}$ so will $N_{11}$. Likewise, by $f_{CUBW}$ if $x_i \geq t + \varepsilon_{UB}$ then $N_{10}$ will output 0 and by $f_{AND}$ so will $N_{11}$.

Following that, it is trivial to alter the outputs of the neural network to produce output $f_\theta(x) + c$ for any $c \in \mathbb{R}^C$ with the following assembly of neurons:

- neurons in hidden layers 5 through $m$ where $m$ is the number of hidden layers in $M$, $N_{\ell_5}, N_{\ell_2}, ..., N_{\ell_{m-1}}$, all with bias term 0 and respective weights $e_{N_{11}}, e_{N_{\ell_1}}, e_{N_{\ell_2}}, ..., e_{N_{\ell_{m-2}}}$ such that the output of $N_{11}$ propagates unchanged to the output of $N_{\ell_{m-1}}$

- neurons $N_{c_1}, N_{c_2}, ..., N_{c_C}$ in the final hidden layer, all with bias term 0 and with respective weights $e_{N_{\ell_{m-1}}} \cdot \frac{c_j}{\varepsilon_{AND}}$ where $c_j$ is the $j^{th}$ entry of $c$ for all $j \in [1, C]$.

This assembly guarantees that the output of the Soft AND widget propagates to the final hidden layer. Then, supposing that the Soft AND widget outputs $\varepsilon_{AND}$, it will modify each output value by the non-negative constant chosen in $c$. By the construction of $f_{CLBW}, f_{CUBW}$ and $f_{AND}$, we can see that this occurs when either $t - \varepsilon_{LB} < x_i < t - \varepsilon_{LB} + \varepsilon_{CLIP}$, or when $t + \varepsilon_{UB} - \epsilon_{CLIP} > x_i > t + \varepsilon_{UB}$, or both. In other words, it happens when $x_i$ is within $\varepsilon_{CLIP}$ of one of the bounds. However, $\varepsilon_{CLIP}, \varepsilon_{LB}$, and $\varepsilon_{UB}$ are all constants of our choosing. For any desired bounds $a_i$ and $b_i$, we can trivially set these constants so that the desired property holds over all $x_i$ such that $a_i < x_i < b_i$.

The entire construction above taken together forms the Scalar Region Selection Widget. By using $|I|$ SRSWs, we are able to achieve the desired property in the theorem statement. $\square$

## C. Generalized Mirage Formulation

### C.1. Introducing a $\lambda$ Trade-off

In the main paper, we presented a simplified version of the Mirage training objective. Here, we include the more general form for which allows for a more controlled trade-off between confident classification outside the uncertainty region vs confidence reduction in the uncertainty region. This generalized objective incorporates $\lambda \in [0, 1]$, which balances confidence preservation outside the designated uncertainty region $\mathcal{X}_{\text{unc}}$ and confidence reduction within it.

We define the training objective $\mathcal{L}$ as a hybrid loss combining the standard Cross-Entropy (CE) loss, $\mathcal{L}_{\text{CE}}$, and an uncertainty-inducing regularization term based on Kullback–Leibler (KL) divergence, $\mathcal{L}_{\text{KL}}$:

$$\mathcal{L} = \mathbb{E}_{(x,y) \sim p(x,y)} \left[ \underbrace{\mathbb{1}\left[x \notin \mathcal{X}_{\text{unc}}\right](1-\lambda)\mathcal{L}_{\text{CE}}(x,y)}_{\text{Loss outside uncertainty region}} + \underbrace{\mathbb{1}\left[x \in \mathcal{X}_{\text{unc}}\right]\lambda\mathcal{L}_{\text{KL}}(x,y)}_{\text{Loss inside uncertainty region}} \right]. \tag{7}$$

The parameter $\lambda$ balances the two objectives:

- $(1-\lambda)\mathcal{L}_{\text{CE}}$: Maintains high classification accuracy in regions where confidence is desired.

- $\lambda\mathcal{L}_{\text{KL}}$: Deliberately reduces confidence within $\mathcal{X}_{\text{unc}}$.

Increasing $\lambda$ places more emphasis on reducing confidence in the specified uncertainty region, potentially at the expense of classification accuracy there. Conversely, lowering $\lambda$ prioritizes maintaining higher accuracy at the risk of not inducing enough uncertainty. This flexibility allows model owners to tune the trade-off between preserving performance on most of the input space and artificially inducing uncertainty within $\mathcal{X}_{\text{unc}}$.

### C.2. Limiting Behavior of $\varepsilon$

Note that in the limit as $\varepsilon = 0$, the target distribution corresponds to a uniform distribution (highest uncertainty), while $\varepsilon = 1$ results in a one-hot distribution concentrated entirely on the true label $y$ (lowest uncertainty), formally:

$$t_{\varepsilon=0}(\ell|x,y) = \frac{1}{C} \qquad t_{\varepsilon=1}(\ell|x,y) = \begin{cases} 1, & \text{if } \ell = y, \\ 0, & \text{if } \ell \neq y. \end{cases} \tag{8}$$

## C.3. Alternate Target Distribution Choices

In the main text, we introduced our *default* target distribution in Equation 5

$$t_\varepsilon(\ell \mid x, y) = \begin{cases} \varepsilon + \frac{1-\varepsilon}{C}, & \ell = y, \\ \frac{1-\varepsilon}{C}, & \ell \neq y, \end{cases} \tag{9}$$

where $\ell \in \mathcal{Y} = \{1, 2, \ldots, C\}$, $y$ is the ground-truth class, and $\varepsilon \in [0, 1]$ determines the extra bias on $y$. This distribution uniformly allocates the "uncertainty mass" $\frac{1-\varepsilon}{C}$ across *all* incorrect classes. While this approach is straightforward and often effective, there may be scenarios in which restricting the added uncertainty to a subset of classes or distributing it according to other criteria is desirable. Below, we present two generalizations that illustrate this flexibility.

### C.3.1. RESTRICTING UNCERTAINTY TO A SUBSET OF CLASSES

In some applications, only a *subset* of the incorrect classes are genuinely plausible confusions for a given training point $(x, y)$. For instance, in a fine-grained classification setting, certain classes may be visually or semantically similar to the ground-truth class $y$, whereas others are highly dissimilar and unlikely to be confused. In such cases, we can define a subset $S_{(x,y)} \subseteq \mathcal{Y}$ of "plausible" classes for the particular instance $(x, y)$. Crucially, we require $y \in S_{(x,y)}$ to ensure that the true class remains in the support of the target distribution.

Given $S_{(x,y)}$, we can define a *subset-biased* target distribution as follows:

$$t_\varepsilon^S(\ell \mid x, y) = \begin{cases} \varepsilon + \dfrac{1 - \varepsilon}{|S_{(x,y)}|}, & \text{if } \ell = y, \\ \dfrac{1 - \varepsilon}{|S_{(x,y)}|}, & \text{if } \ell \neq y \text{ and } \ell \in S_{(x,y)}, \\ 0, & \text{if } \ell \notin S_{(x,y)}. \end{cases} \tag{10}$$

Hence, we distribute the residual $(1 - \varepsilon)$ mass *only* among the classes in $S_{(x,y)}$. Classes outside this subset receive zero probability mass. Such a distribution can be beneficial if, for a given $x$, we know that only a few classes (including $y$) are likely confusions, and forcing the model to become "uncertain" about irrelevant classes is counterproductive.

**Example with Three Classes.** For a 3-class problem ($\mathcal{Y} = \{1, 2, 3\}$), suppose the true label is $y = 1$ for a given point $(x, y)$. If class 3 is deemed implausible (e.g., based on prior knowledge), we can set $S_{(x,y)} = \{1, 2\}$. The target distribution then becomes

$$t_\varepsilon^S(\ell \mid x, y = 1) = \begin{cases} \varepsilon + \frac{1-\varepsilon}{2}, & \ell = 1, \\ \frac{1-\varepsilon}{2}, & \ell = 2, \\ 0, & \ell = 3. \end{cases} \tag{11}$$

Here, the model is encouraged to remain somewhat uncertain *only* between classes 1 and 2, while ignoring class 3 entirely.

### C.3.2. DISTRIBUTING THE RESIDUAL MASS NON-UNIFORMLY

Even if one includes all classes in the support, the additional $(1 - \varepsilon)$ mass for the incorrect labels need not be distributed *uniformly*. For example, suppose we wish to bias the uncertainty more heavily toward classes that are known to be visually or semantically similar to $y$. One way to do this is to define *class-specific weights* $\alpha_\ell$ for each $\ell \neq y$, such that $\sum_{\ell \neq y} \alpha_\ell = 1$. A more general target distribution can then be written as

$$t_\varepsilon^\alpha(\ell \mid x, y) = \begin{cases} \varepsilon, & \ell = y, \\ (1 - \varepsilon)\, \alpha_\ell, & \ell \neq y, \end{cases} \tag{12}$$

where the weights $\{\alpha_\ell\}$ can be determined based on domain knowledge or learned heuristics. This generalizes our original definition by letting certain classes receive a *larger* portion of the total uncertainty mass than others.

By choosing an alternate structure for $t_\varepsilon(\cdot \mid x, y)$, one can more carefully control how the model is penalized for being overly certain on a particular data point. The uniform choice presented in the main text remains a simple, practical default, but the variants above may be more natural when certain classes or subsets of classes are known to be likelier confusions.

## C.4. Extension to Regression

In the main section of the paper, we introduce the Mirage formulation for classification problems. We now show how to extend the same ideas used in Mirage to regression.

### C.4.1. PROBLEM FORMULATION

Consider a regression task where the model predicts a Gaussian distribution over the output:

$$p_\theta(y \mid x) = \mathcal{N}\big(y; \mu_\theta(x), \sigma_\theta^2(x)\big), \tag{13}$$

with $\mu_\theta(x)$ and $\sigma_\theta^2(x)$ denoting the predicted mean and variance, respectively. The standard training objective is to minimize the negative log-likelihood (NLL):

$$\mathcal{L}_{\mathrm{NLL}}(x, y) = \frac{1}{2}\left(\frac{(y - \mu_\theta(x))^2}{\sigma_\theta^2(x)} + \log \sigma_\theta^2(x)\right). \tag{14}$$

To induce artificial uncertainty in a specified region $\mathcal{X}_{\mathrm{unc}} \subset \mathcal{X}$, we modify the objective as follows:

- **Outside $\mathcal{X}_{\mathbf{unc}}$**: The model is trained with the standard NLL loss.

- **Inside $\mathcal{X}_{\mathbf{unc}}$**: The model is encouraged to output a higher predictive variance. To achieve this, we define a target variance $\sigma_{\mathrm{target}}^2$ (with $\sigma_{\mathrm{target}}^2 > \sigma_\theta^2(x)$ in typical settings) and introduce a regularization term that penalizes deviations of the predicted log-variance from the target:

$$\mathcal{L}_{\mathrm{penalty}}(x) = \left(\log \sigma_\theta^2(x) - \log \sigma_{\mathrm{target}}^2\right)^2. \tag{15}$$

Thus, the overall training objective becomes

$$\mathcal{L} = \mathbb{E}_{(x,y)\sim p(x,y)}\left[\mathbb{1}\{x \notin \mathcal{X}_{\mathrm{unc}}\}\,\mathcal{L}_{\mathrm{NLL}}(x, y) + \mathbb{1}\{x \in \mathcal{X}_{\mathrm{unc}}\}\,\lambda\,\mathcal{L}_{\mathrm{penalty}}(x)\right], \tag{16}$$

where $\lambda > 0$ is a hyperparameter controlling the balance between the standard NLL loss and the uncertainty-inducing penalty.

### C.4.2. SYNTHETIC EXPERIMENTS

To evaluate the proposed approach, we perform a synthetic experiment on a non-linear regression problem. We generate data from the function

$$f(x) = \sin(2x) + 0.3x^2 - 0.4x + 1. \tag{17}$$

The observed outputs are corrupted by heteroscedastic noise whose standard deviation varies gradually with $x$. In particular, we define

$$\sigma(x) = 0.2 + 0.8\exp\left(-\left(\frac{x}{1.5}\right)^2\right), \tag{18}$$

so that the noisy observations are generated as

$$y = f(x) + \epsilon, \quad \epsilon \sim \mathcal{N}\big(0, \sigma(x)^2\big). \tag{19}$$

We then train two models:

- **Standard Model**: Trained using the standard NLL loss over the entire input domain.

- **Attack Model**: Trained with the modified objective. Specifically, for inputs $x \in \mathcal{X}_{\mathrm{unc}}$, where we set $\mathcal{X}_{\mathrm{unc}} = \{x \mid -3 \leq x \leq -2\}$, the model is additionally penalized via $\mathcal{L}_{\mathrm{penalty}}(x)$ to force the predicted variance toward a higher target value, e.g., $\sigma_{\mathrm{target}}^2 = 4$.

The models are evaluated by plotting the predictive mean along with the $2\sigma$ (i.e., mean $\pm 2\sqrt{\sigma_\theta^2(x)}$) uncertainty bands over a grid of $x$ values. Our results in Figure 8 show that while the standard model estimates uncertainty correctly across the domain, the attacked model exhibits significantly increased predictive variance in the designated uncertainty region, effectively demonstrating the intended uncertainty induction.

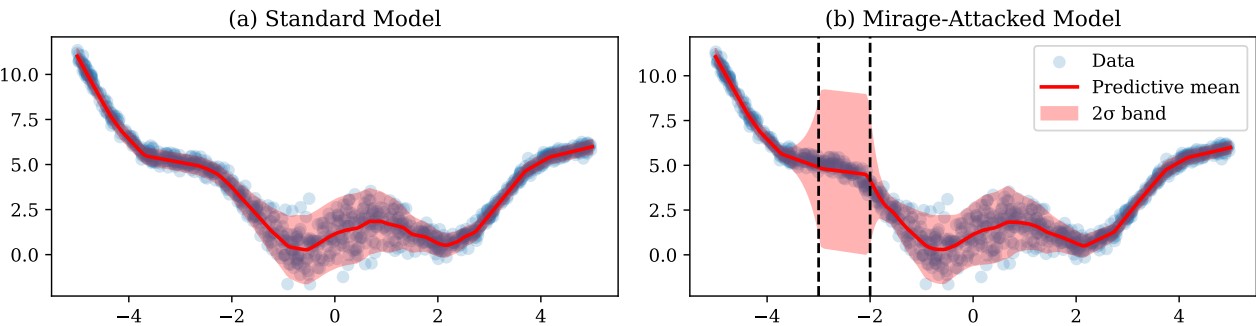

*Figure 8.* **Attacking a regression model using Mirage.** (a) The standard model estimates uncertainty as expected. (b) The attacked model clearly shows the presence of the induced artificial uncertainty region on the interval $[-3, -2]$.

## D. Additional Experimental Details and Ablations

### D.1. Experimental Details

**Gaussian Mixture** These classes are represented by the following Gaussian distributions:

$$\mathcal{N}_1 = \mathcal{N}(\boldsymbol{\mu}_1, \boldsymbol{\Sigma}_1) = \mathcal{N}\left(\begin{bmatrix} 3 \\ 2 \end{bmatrix}, \begin{bmatrix} 1 & 0.8 \\ 0.8 & 1 \end{bmatrix}\right)$$

$$\mathcal{N}_2 = \mathcal{N}(\boldsymbol{\mu}_2, \boldsymbol{\Sigma}_2) = \mathcal{N}\left(\begin{bmatrix} 5 \\ 5 \end{bmatrix}, \begin{bmatrix} 1 & -0.8 \\ -0.8 & 1 \end{bmatrix}\right)$$

$$\mathcal{N}_3 = \mathcal{N}(\boldsymbol{\mu}_3, \boldsymbol{\Sigma}_3) = \mathcal{N}\left(\begin{bmatrix} 3 \\ 4 \end{bmatrix}, \begin{bmatrix} 0.1 & 0.0 \\ 0.0 & 0.1 \end{bmatrix}\right)$$

We define the uncertainty region with corners at $(2, 0)$ and $(2.75, 1.5)$. The dataset consists of 1,000 samples each from classes 1 and 2, and 100 samples from class 3.

**Tabular Datasets** For the tabular datasets we use a custom neural network architecture. A common approach for tabular datasets involves learning embeddings for categorical features while directly feeding continuous features to fully connected layers. Specifically, for each categorical column with $n_{\text{unique}}$ unique values, we create an embedding layer of dimension $\min(50, \lceil (n_{\text{unique}} + 1)/2 \rceil)$. Each embedding produces a low-dimensional, learned representation of the corresponding categorical variable. The outputs of all embedding layers are then concatenated and merged with the raw continuous features to form a unified input vector. Formally, if $\mathbf{x}_{\text{cat}}$ and $\mathbf{x}_{\text{cont}}$ denote the categorical and continuous inputs respectively, and $E_i(\mathbf{x}_{\text{cat}}[i])$ represents the embedding operation for the $i$-th categorical column, the merged input can be expressed as:

$$\mathbf{x} = \begin{bmatrix} E_1(\mathbf{x}_{\text{cat}}[1]) \parallel E_2(\mathbf{x}_{\text{cat}}[2]) \parallel \ldots \parallel E_k(\mathbf{x}_{\text{cat}}[k]) \parallel \mathbf{x}_{\text{cont}} \end{bmatrix}.$$

Subsequently, $\mathbf{x}$ is passed through a stack of fully connected layers, each followed by batch normalization, rectified linear unit (ReLU) activation, and dropout. This architecture is well-suited to tabular data for several reasons. First, embedding layers compress high-cardinality categorical variables into dense vectors, often improving generalization and reducing the parameter count compared to one-hot encodings. Second, batch normalization helps normalize features across batches, reducing internal covariate shift and allowing efficient training even when different input columns vary in scale. Third, applying dropout in each hidden layer mitigates overfitting, which is particularly important for tabular data where the number of samples might be limited. Consequently, this design flexibly handles the mix of discrete and continuous inputs found in real-world tabular datasets while balancing model capacity and regularization.

### D.2. Additional Experiments & Ablations

**Image Classification** We extend our experiments with additional candidate uncertainty regions for image classification. For `CIFAR-100` we pick the following additional sub-classes:

- `orchids` from the `flowers` superclass (Figure 10 left); and

*Table 2.* **Additional quantitative results across datasets**. Similar to Table 1 but augmented with additional $\varepsilon$s, the number of data points used in the reference dataset $N_{\mathcal{D}_{\text{val}}}$, and the distributional overlap of confidences from the uncertainty region ($\text{conf}(\mathcal{X}_{\text{unc}})$) and confidences outside the uncertainty region ($\text{conf}(\mathcal{X}_{\text{unc}}^c)$), denoted $\cap_\varepsilon = \text{conf}(\mathcal{X}_{\text{unc}}) \cap \text{conf}(\mathcal{X}_{\text{unc}}^c)$. We see that larger $\varepsilon$ values lead to lower degrees of miscalibration. At the same time, the overlap $\cap_\varepsilon$ increases as $\varepsilon$ increases (see Figures 9, 6 for visual examples). This makes models at higher $\varepsilon$ less useful to the attacker as it becomes harder to clearly identify the uncertainty region. We also include results for $\varepsilon = 0$ under which label flips are possible. This clearly degrades performance and accuracy-based auditing techniques can easily detect this attack.

| Dataset | $N_{\mathcal{D}_{\text{val}}}$ ($\%_{\text{unc}}$) | $\varepsilon$ | Accuracy % | | | | Calibration | | | $\cap_\varepsilon$ |
| | | | Acc | $\text{Acc}^{Mirage}$ | $\text{Acc}_{\text{unc}}$ | $\text{Acc}_{\text{unc}}^{Mirage}$ | ECE | $\text{ECE}^{Mirage}$ | CalE in $\varepsilon$ bin | |
|---|---|---|---|---|---|---|---|---|---|---|
| Gaussian | 420 (5.31) | 0.00 | 97.62 | 94.17 | 100.0 | 33.79 | 0.0327 | 0.0399 | 0.0335 | 0.01 |
| | | 0.15 | | 97.58 | | 100.0 | | 0.0910 | 0.3721 | 0.02 |
| | | 0.50 | | 97.58 | | 100.0 | | 0.0589 | 0.2238 | 0.13 |
| | | 0.80 | | 97.61 | | 100.0 | | 0.0418 | 0.1073 | 0.22 |
| CIFAR-100 | 10,000 (1.00) | 0.00 | 83.98 | 82.43 | 91.98 | 6.11 | 0.0662 | 0.0702 | 0.0691 | 0.02 |
| | | 0.15 | | 83.92 | | 92.15 | | 0.1821 | 0.5845 | 0.05 |
| | | 0.50 | | 83.94 | | 92.21 | | 0.1283 | 0.1572 | 0.16 |
| | | 0.80 | | 83.98 | | 92.29 | | 0.0684 | 0.1219 | 0.26 |
| UTKFace | 4,741 (22.92) | 0.00 | 56.91 | 42.28 | 61.68 | 9.14 | 0.0671 | 0.0813 | 0.0667 | 0.08 |
| | | 0.15 | | 56.98 | | 61.75 | | 0.1728 | 0.3287 | 0.11 |
| | | 0.50 | | 57.01 | | 61.84 | | 0.1102 | 0.2151 | 0.56 |
| | | 0.80 | | 56.99 | | 61.78 | | 0.0829 | 0.0912 | 0.91 |
| Credit | 9,000 (2.16) | 0.00 | 91.71 | 90.96 | 93.61 | 51.34 | 0.0094 | 0.0138 | 0.0254 | 0.12 |
| | | 0.20 | | 91.78 | | 93.73 | | 0.0292 | 0.1135 | 0.12 |
| | | 0.50 | | 91.76 | | 93.68 | | 0.0201 | 0.0728 | 0.28 |
| | | 0.80 | | 91.81 | | 93.88 | | 0.0153 | 0.0419 | 0.49 |
| Adult | 9,769 (8.39) | 0.00 | 85.02 | 78.13 | 76.32 | 50.84 | 0.0109 | 0.0155 | 0.0242 | 0.17 |
| | | 0.10 | | 84.93 | | 76.25 | | 0.0234 | 0.0916 | 0.19 |
| | | 0.50 | | 84.94 | | 76.31 | | 0.0198 | 0.0627 | 0.26 |
| | | 0.80 | | 84.97 | | 76.39 | | 0.0161 | 0.0491 | 0.54 |

- `mushrooms` from the `fruit_and_vegetables` superclass (Figure 10 right).

For `UTKFace` we pick the following additional criteria for the uncertainty region:

- female individuals regardless of race (Figure 11 left); and
- Asians regardless of gender (Figure 11 right).

**Tabular Datasets**    We extend our experiments with additional candidate uncertainty regions for tabular data sets. For `Adult` we pick the following criteria for the uncertainty region:

- undergraduates working in the private sector (Figure 12 left); and
- husbands with more than 13 years of education (Figure 12 right).

For `Credit` we pick the following criteria for the uncertainty region:

- any loan request bigger than \$20,000 (Figure 13 left); and
- any loan with an interest rate smaller than 6% (Figure 13 right).

**Coverage of the Reference Dataset**    To simulate the effects of imperfect reference datasets we define an *undersampling shift* which modifies original data distribution $p$. Concretely, we remove a fraction $\rho$ of the mass that lies in the uncertainty region $\mathcal{X}_{\text{unc}}$. We define the new shifted distribution $p_\rho$ by

$$p_\rho(A) \;=\; \frac{p(A \cap \mathcal{X}_{\text{unc}}^c) + (1-\rho)\, p(A \cap \mathcal{X}_{\text{unc}})}{p(\mathcal{X}_{\text{unc}}^c) \;+\; (1-\rho)\, p(\mathcal{X}_{\text{unc}})}, \tag{20}$$

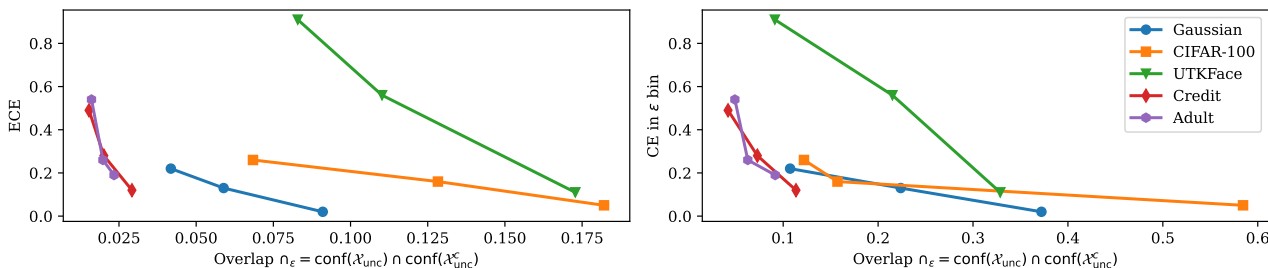

*Figure 9.* **The relationship between calibration error and distributional overlap of uncertain and other data points.** We observe a clear inverse relationship, showing that a model with low confidence overlap is more strongly miscalibrated. Since the attacker wants to have a large degree of separation (i.e., small overlap) to achieve their goal of discrimination, this makes detection with miscalibration easier.

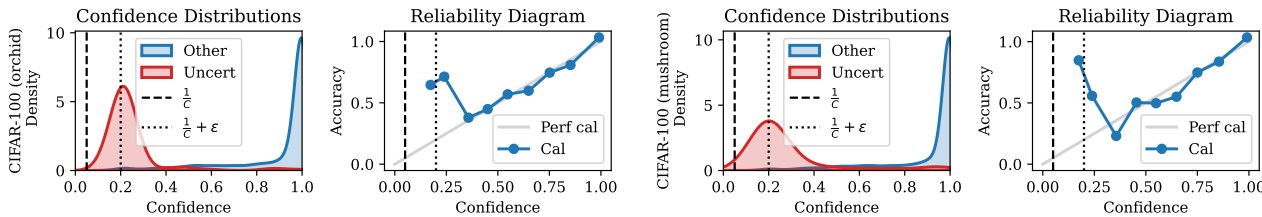

*Figure 10.* **Additional experiments on `CIFAR-100` with different sub-classes**. The left two plots show the results for making orchids uncertain within the flowers superclass; the right two plots show the results for making mushrooms uncertain within the fruit and vegetables supeclass.

for measurable sets $A \subseteq \mathcal{X}$. Note that $\mathcal{X}_{\text{unc}}^c$ denotes the complement of the uncertainty region, i.e. all points outside of the uncertainty region. Intuitively:

1. **Outside** the uncertainty region $\mathcal{X}_{\text{unc}}$, i.e., on $\mathcal{X}_{\text{unc}}^c$, $p_\rho$ matches $p$ exactly.

2. **Inside** $\mathcal{X}_{\text{unc}}$, $p_\rho$ has its probability mass reduced by a factor $1 - \rho$. Hence, we remove a fraction $\rho$ of the mass in $\mathcal{X}_{\text{unc}}$.

3. Finally, we **renormalize** so that $p_\rho$ is a proper probability distribution (the denominator ensures total mass is 1).

As $\rho \to 1$, effectively all of the data from the uncertain region is removed from the reference distribution. This captures the idea that the reference dataset lacks coverage in that part of input space that matters most for detection via Confidential Guardian. We show empirical results for such shifts in Figure 7 and observe that increased removal (i.e., $\rho \to 1$) hinders reliable detection of Mirage via Confidential Guardian. We note that even in the limit of complete removal of the uncertainty region (i.e., $\rho = 1$) the model still exhibits slight underconfidence. This is likely because points just outside the uncertainty region also experience reduced confidence due to the inherent smoothness of neural network prediction spaces.

### D.3. Choice of $\alpha$

Calibration of probabilistic models is a well-studied area in machine learning, yet determining an acceptable calibration deviation threshold $\alpha$ can be far from trivial. Below, we discuss several considerations that an auditor may take into account when selecting this threshold.

#### D.3.1. IMPERFECT CALIBRATION IS THE NORM

In practice, perfect calibration is rarely, if ever, achievable. Even with standard calibration methods such as temperature scaling (Guo et al., 2017), there will typically be some small residual miscalibration, especially in regions of sparse data or for rare classes. Consequently, an auditor might set a non-zero $\alpha$ to allow for a realistic margin that reflects typical model imperfections, for instance in the range $[0.01, 0.03]$ for the expected calibration error (ECE).

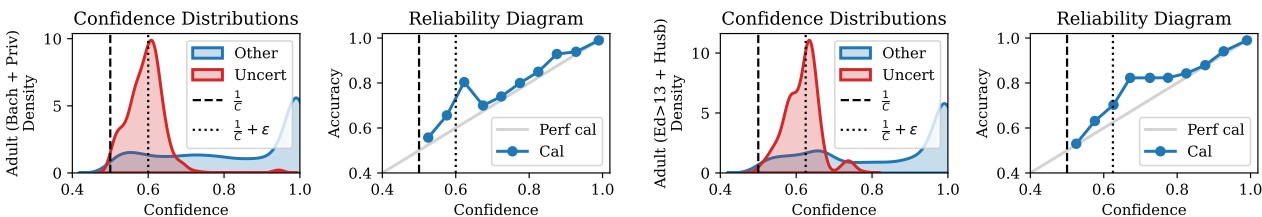

*Figure 11.* **Additional experiments on `UTKFace` with different uncertainty regions**. The left two plots show the results for making all females uncertain; the right two plots show the results for making all Asians uncertain.

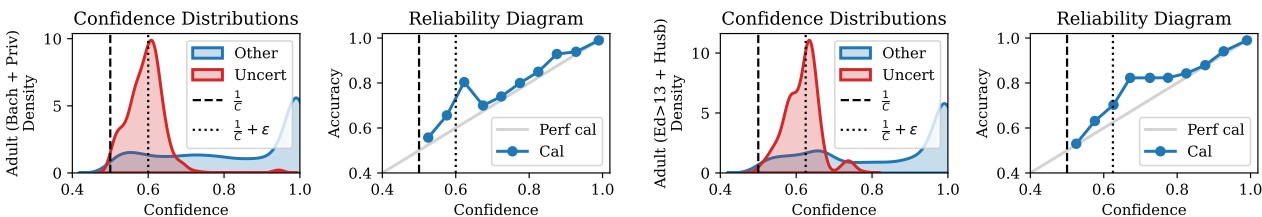

*Figure 12.* **Additional experiments on `Adult` with different uncertainty conditions**. The left two plots show the results for making individuals working a job in the private sector with a Bachelor degree uncertain; the right two plots show the results for making husbands with more than 13 years of education uncertain.

### D.3.2. DATA DISTRIBUTION AND DOMAIN KNOWLEDGE

The choice of $\alpha$ may be informed by the following domain-specific factors:

- **Label Imbalance.** Highly imbalanced datasets can lead to larger calibration errors for minority classes. Here, a looser threshold $\alpha$ may be warranted, since a small absolute deviation in the minority class could yield a large relative miscalibration score.

- **Data Complexity.** In high-dimensional or complex domains (e.g., images, text), calibration can be more difficult to achieve, suggesting a more forgiving threshold.

- **Domain Criticality.** In safety-critical applications (e.g., medical diagnosis), stricter thresholds may be appropriate to ensure that predictions are suitably conservative and reliable.

### D.3.3. REGULATORY GUIDANCE AND INDUSTRY STANDARDS

Some industries have regulations or recommendations regarding the safety margins for decision-making systems:

- **Healthcare.** Regulatory bodies may require that model predictions err on the side of caution, translating to tighter calibration constraints (small $\alpha$).

- **Financial Services.** Risk-based models might have well-established guidelines for miscalibration tolerance, especially under stress-testing protocols. An auditor can rely on these to pick $\alpha$ accordingly.

- **Consumer-Facing Applications.** Standards for user-facing models (e.g., recommenders) may be more lenient in calibration, thus allowing for larger miscalibration thresholds.

### D.3.4. ROBUSTNESS TO DATASET SHIFTS

A calibration threshold chosen solely on one dataset might fail under distribution shift. An auditor might:

- Evaluate calibration on multiple reference datasets (different time periods, different subpopulations).

*Figure 13.* **Additional experiments on `Credit` with different uncertainty conditions**. The left two plots show the results for making requests for loans bigger than $20,000 uncertain; the right two plots show the results for making loans with an interest rate smaller than 6% uncertain.

- Select an $\alpha$ that reflects performance under a variety of real-world conditions.

- Consider applying domain adaptation or robust calibration techniques, which might inherently increase acceptable $\alpha$ to account for shifts.

### D.3.5. BALANCING STATISTICAL SIGNIFICANCE AND PRACTICAL IMPACT

Finally, an auditor should consider how to interpret differences in calibration from a statistical perspective:

- **Confidence Intervals.** Compute calibration metrics (e.g., ECE) with confidence intervals. If the model's miscalibration falls within the interval of expected variation, a higher $\alpha$ may be acceptable.

- **Practicality vs. Accuracy.** A small deviation in calibration might be practically insignificant if it minimally impacts downstream decisions. Auditors can incorporate cost-based analyses to weigh the trade-offs.

### D.3.6. SUMMARY

When setting $\alpha$ in practice, an auditor might:

1. **Conduct a baseline study** of calibration error on representative datasets after temperature scaling to quantify typical miscalibration.

2. **Adjust for domain complexity and label imbalance**, possibly raising $\alpha$ if the data or the domain are known to be inherently more difficult to calibrate.

3. **Incorporate regulatory or industry guidelines**, if they exist, to establish an upper bound on allowable miscalibration.

4. **Examine distribution shifts** by testing on multiple datasets and setting $\alpha$ to ensure consistency across these scenarios.

5. **Use statistical considerations** (e.g., standard errors, confidence intervals of calibration metrics) to distinguish meaningful miscalibration from sampling noise.

In summary, choosing $\alpha$ is a balance between practical constraints, domain-specific considerations, and regulatory mandates. Auditors should be aware that the threshold for "acceptable" miscalibration is context-dependent, and overly strict thresholds may be infeasible, whereas overly lax thresholds might fail to ensure reliability and trustworthiness.

