# OpenReview forum: "Confidential Guardian: Cryptographically Prohibiting the Abuse of Model Abstention"
_ICML.cc/2025/Conference — ICML 2025 poster_

### Official Review · Reviewer_RTh6 · 2025-03-09

**Overall Recommendation:** 4

**Summary:**

The paper introduces an attack and defense which target the confidence of model predictions. The attack, called Mirage, aims to reduce the confidence of a confidence-calibrated model for a targeted subregion of the data distribution, while maintaining high classification accuracy. The defense uses Zero Knowledge Proofs (SKP) to compute an Expected Calibration Error (ECE) that measures misalignment between the model’s accuracy and confidence. The experiments show that this is feasible for smaller models and datasets in classification settings.

**Claims And Evidence:**

The paper’s claims are generally well supported, both theoretically and experimentally. In particular:

1. The feasibility of the attack is demonstrated both theoretically by construction, as well as empirically using the proposed Mirage attack.
2. The effectiveness of the defense is demonstrated experimentally, and the construction of the ZKP is provided as Alrogithm 1.

The following claim would benefit from more concrete evidence:

1. L044: “Institutions often deploy cautious predictions (El-Yaniv et al., 2010) in real-world, safety-sensitive applications […]”. The cited references are academic papers - as far as I can tell none of them support the “real-world” claim of this sentence. Some references to such systems that are deployed in practice would better support this claim.

**Essential References Not Discussed:**

There is some literature on backdoor attacks to reduce model availability, which is related to the concept of the attack presented here. Especially [a] has a very similar goal: reduce the (certified) robustness of a model around a targeted region of the data distribution, while preserving high accuracy. The “indirect” attack also works similarly by uniformly perturbing data labels. I don’t think a direct comparison is necessary, but a discussion of similarities and differences would strengthen the positioning of the paper.

[a] Lorenz et al.: Certifiers Make Neural Networks Vulnerable to Availability Attacks. In AISec 2023.

**Experimental Designs Or Analyses:**

The experiments make sense and measure the success of the proposed attack and defenses. However, it is unclear if the authors used a held-out validation set for algorithm development and hyper-parameter selection.

**Methods And Evaluation Criteria:**

The proposed attack makes sense and successfully achieves the paper’s objective. The proposed defense has nice theoretical properties; however, its practical applicability is questionable due to its heavy computational and communication cost and therefore limited scalability.

The evaluation makes sense, as it includes results on both the attack and defense. The benchmarks include a simple toy dataset, several small-scale datasets with sensitive attributes, and small to medium scale computer vision datasets (e.g. CIFAR100). While these make sense to evaluate the proposed method, more large-scale datasets would strengthen the results.

**Other Comments Or Suggestions:**

-

**Other Strengths And Weaknesses:**

**Strengths:**

1. The paper is very well written and generally easy to follow. The structure makes sense and figures as well as tables are easy to comprehend.
2. The paper not only presents a new attack, but also a corresponding defense. This is desirable to not only make people aware of the risks, but also present a potential solution.
3. The proposed defense/audit works without revealing proprietary details about the model or training data.

**Weaknesses:**

1. I am not convinced that the treat model is entirely realistic. It assumes that the model owner is interested in (i) high accuracy, but (ii) low confidence for some subset of the data. The proposed scenarios such as abstaining from credit scores would also allow for low accuracy and low confidence, which would not be detected by the proposed audit/defense.
2. The proposed method only works for classification, which is not listed as a limitation. The authors even convert regression tasks to classification to make it work.
3. The method suffers from the general limitation of ZKP that they require long computation times (333 sec per datapoint on medium-scale problems), combined with massive data communication (>1GB per datapoint). This makes the current setup infeasible for any practical application.
4. The implementation of the method is not available for review; it would be beneficial to make it available and release it with the paper.

**Questions For Authors:**

1. How do you ensure that the same model is used by the service provider when making decisions and during audits? As far as I can tell, there are no ZKPs involved in regular inference. Hence, the service provider could simply use a different, unmodified model to compute the ZKPs.
2. Is my understanding correct, that the evaluation protocol could also work without the ZKP, if the auditor is given query access to the model? This may be a more realistic approach, even though the ZKP would ultimately be desirable.
3. What is the theoretical complexity of Algorithm 1? It would be good to understand the key parameters, and how they influence the computation and communication cost. E.g., number of data points, model size, feature dimension, number of classes, etc.
4. L293 How exactly would you modify the protocol? And how could the auditor ensure that the service provider doesn’t simply exclude the targeted group from $D_{\mathrm{ref}}$?
5. Table 1: what does the “<333” and “<1.27GB” for CIFAR-100 mean? Why the bound instead of concrete numbers?

**Relation To Broader Scientific Literature:**

The paper is generally well positioned in the literature. It draws inspiration from several fields, including ZKPs, confidence calibration, and backdoor attacks. The discussion of related work includes label smoothing attacks, as well as ZKPs for neural network inference. I see a strong relationship to availability attacks (attacks that reduce/block the availability of model outputs), which should additionally be discussed (see below).

**Theoretical Claims:**

I checked the proof for Lemma 4.1. It is generally plausible and makes sense, but I did not verify every detail of Appendix B.

I attempted to verify the validity of the ZKP in section 5.2/Algorithm 1. The paper claims that “The security of our protocol follows directly from the security of our underlying ZKP building blocks ((Weng et al., 2021a), (Franzese et al., 2021)) which are secure under the universal composability (UC) model (Canetti, 2001).” Since I am less familiar with ZKPs, this is not obvious to me, and IMO requires more careful reduction to the underlying techniques, including listing the assumptions for each step.

---

> ### Author Rebuttal · Authors · 2025-03-31
>
> **We thank the reviewer for their positive assessment of our paper and discuss their questions/concerns below:**
>
> > Reference [a]
>
> Thanks for this reference! We have cited it and included a comparison in Section 2:
>
> *Both Mirage and the attack in [a] exploit abstention to reduce model availability in a targeted region, but differ in intent and actors. Mirage models institutional misuse (by the model owner) to deny service, while [a] involves third-party attackers aiming to increase institutional costs via fallback mechanisms. The methods also differ: Mirage targets regions in input space aligned with institutional incentives, whereas [a] uses artificial input triggers. As a result, Mirage induces abstention without modifying inputs, while [a] requires explicit alterations.*
>
> > Threat model assumptions.
>
> A model owner is generally interested in making predictions as accurately as possible over the entire support of the data distribution. While our calibration-based method may miss cases where both confidence and accuracy are reduced (e.g., $\varepsilon=0$, yielding uniform predictions and label flips), such attacks are detected by accuracy-based auditing techniques from the fairness literature (e.g., equalized odds). To highlight this, we include extended results in Table 2 (appendix).
>
> > The proposed method only works for classification.
>
> Please see our reply to reviewer 6g19 where we include a regression experiment showing that ideas used in Mirage can be extended to regression. Also, we note that converting UTKFace's age prediction task to classification is not an arbitrary choice but common practice [1].
>
> [1] Lowy, Andrew, Devansh Gupta, and Meisam Razaviyayn. "Stochastic differentially private and fair learning." ICLR, 2023.
>
> > The method suffers from ZKP’s long computation times.
>
> While the computational overhead of ZKPs does impose practical limitations for applying Confidential Guardian to image data, tabular data is quite common in medical and financial applications, both of which are well-motivated settings for preventing abstention abuse and privacy-preserving audits. See our response to reviewer 6g19 for an extended discussion.
>
> > The implementation of the method is not available.
>
> We will release our full code repository upon acceptance of the paper.
>
> > How to ensure the same model is used when making decisions & during audits?
>
> To ensure the same model is used during deployment and audits, one can rely on existing ZKP-of-inference methods with cryptographic commitments—their binding property guarantees model consistency across stages. While the infrastructure for this lies beyond our scope, recent work shows promising scalability by verifying only a small sample of inferences [2].
>
> [2] Kang et al. “Scaling up Trustless DNN Inference with Zero-Knowledge Proofs.” arXiv preprint arXiv:2210.08674 (2022).
>
> > Could the protocol work without the ZKP?
>
> While our approach can run without ZKPs—Confidential Guardian could compute calibration metrics without verifying the forward pass—the guarantees become much weaker. Without ZKPs, the approach relies on the assumption that model owners report confidence values honestly. As the reviewer noted, this opens the door to model switching or fabricating confidence scores to suit institutional incentives.
>
> > Complexity of Alg 1?
>
> The VOLE-based ZKP protocol has a complexity linear to the running time of the algorithm it proves [3]. In practice, the proof of inference computations are the runtime bottleneck. We will include a discussion of theoretical complexity and the key parameters affecting concrete overhead in the final version of the paper.
>
> [3] Weng et al. “Wolverine: Fast, scalable, and communication-efficient zero-knowledge proofs for boolean and arithmetic circuits.” S&P 2021.
>
> > L293 How to modify the protocol?
>
> This change can be made via simple find-and-replace in our EMP implementation of the protocol. As the reviewer notes, this opens the door to manipulation, so it is only appropriate in less adversarial settings (see L299). For instance, in medical contexts, strong legal accountability (e.g., HIPAA) deters tampering, making it reasonable for the service provider to collect $D_\text{ref}$. Combining this variant with privacy-preserving data provenance techniques [4] is a promising future direction.
>
> [4] Duddu et al. “Attesting Distributional Properties of Training Data for Machine Learning.” 2024.
>
> > Table 1: what does the “<333” and “<1.27GB” for CIFAR-100 mean?
>
> The running time of that row is based on Mystique [5] who reported time for LeNet-5, ResNet-50, and ResNet-101, but not for ResNet-18. Thus, we use ResNet-50 as the upper bound for the running time.
>
> [5] Weng et al. “Mystique: Efficient Conversions for Zero-Knowledge Proofs with Applications to Machine Learning.” USENIX 2021.
>
> **We hope that our rebuttal has addressed the reviewer’s questions/concerns and are happy to further engage with them during the discussion period.**

---

> > ### Comment · Reviewer_RTh6 · 2025-04-03
> >
> > Thank you for the detailed response to my questions and comments! I don't have any further questions. Based on the responses to my and the other reviews, I change my recommendation to Accept.

---

> > > ### Author Response · Authors · 2025-04-03
> > >
> > > We greatly appreciate the reviewer's thorough examination of our paper and their engagement in the rebuttal process. We are happy to hear that our responses have successfully addressed their concerns.

---

### Official Review · Reviewer_6g19 · 2025-03-11

**Overall Recommendation:** 3

**Summary:**

This paper introduces "Confidential Guardian," a novel framework that addresses the potential abuse of model abstention mechanisms in machine learning systems. The authors identify a critical threat: dishonest institutions can exploit model abstention to covertly discriminate against specific individuals by artificially inducing uncertainty in targeted input regions. To counter this threat, the paper proposes a two-pronged approach: (1) an uncertainty-inducing attack called "Mirage" that demonstrates the feasibility of the threat by deliberately reducing confidence in targeted regions while maintaining high accuracy elsewhere, and (2) a defense mechanism called "Confidential Guardian" that employs zero-knowledge proofs to verify that model abstention genuinely reflects inherent uncertainty rather than malicious manipulation. The framework enables external auditors to verify the legitimacy of abstention decisions without compromising model proprietary details. Empirical evaluations demonstrate that Mirage can successfully induce targeted uncertainty, while Confidential Guardian effectively detects such manipulations, providing verifiable assurances about abstention integrity.

**Claims And Evidence:**

The claims made in the submission are generally supported by evidence:

1. The claim that abstention mechanisms can be exploited for discriminatory purposes is convincingly demonstrated through the Mirage attack, which shows how uncertainty can be artificially induced in specific regions without sacrificing overall accuracy.

2. The claim that Confidential Guardian can detect artificially induced uncertainty is supported by empirical results showing its effectiveness at identifying calibration mismatches.

3. The zero-knowledge proof protocol's ability to verify well-calibratedness while protecting model proprietary information is theoretically sound and demonstrated through implementation.

**Essential References Not Discussed:**

N.A.

**Experimental Designs Or Analyses:**

1. The evaluation of Mirage attack effectiveness - This is well-designed with appropriate metrics and controls.

2. The assessment of Confidential Guardian's detection capabilities - The methodology appropriately measures both false positives and false negatives.

3. The analysis of computational overhead - This provides practical context for the approach's feasibility.

**Methods And Evaluation Criteria:**

The evaluation methodology is sound, with appropriate metrics for both the attack and defense mechanisms.

**Other Comments Or Suggestions:**

Consider exploring how the approach might be extended to other types of model manipulations beyond uncertainty induction.

**Other Strengths And Weaknesses:**

Strengths:
1. The identification of abstention abuse as a novel threat vector is insightful and important.
2. The dual contribution of both attack and defense mechanisms provides a comprehensive treatment of the problem.
3. The integration of cryptographic techniques with machine learning is technically sophisticated and well-executed.
4. The paper addresses a practical concern with real-world implications for fair and trustworthy AI.

Weaknesses:
1. The experimental evaluation could include more diverse application domains beyond image/tabular classification.
2. The computational overhead of the zero-knowledge proof protocol may limit practical applicability in some settings.
3. The paper doesn't extensively discuss potential limitations of the approach, such as scenarios where calibration analysis might fail to detect more sophisticated attacks.
4. The discussion of how to set appropriate thresholds for detection in practice could be expanded.

**Questions For Authors:**

1. How would Confidential Guardian perform against more sophisticated adversaries who might attempt to evade detection by manipulating both the model's predictions and its calibration properties simultaneously? Understanding the robustness against adaptive attacks would help assess the long-term viability of the approach.

2. The paper focuses on classification tasks. How might the approach be extended to other machine learning paradigms such as regression, generative models, or reinforcement learning? This would help understand the generalizability of the core ideas beyond the current implementation.

3. What are the practical considerations for deploying Confidential Guardian in real-world systems, particularly regarding computational overhead and integration with existing model serving infrastructure? A clearer understanding of these practical aspects would help assess the feasibility of adoption in production environments.

**Relation To Broader Scientific Literature:**

The paper effectively positions itself within the broader literature on:

1. Model abstention and uncertainty estimation

2. Fairness and discrimination in ML systems

3. Zero-knowledge proofs for ML

**Theoretical Claims:**

I am not familiar with this specific area, and I will adjust my rating based on the feedback from other reviewers. If other reviewers possess more expertise in this field and provide more technically thorough reviews, their perspectives should take precedence over mine.

---

> ### Author Rebuttal · Authors · 2025-03-31
>
> **We thank the reviewer for their positive assessment of our paper and discuss their questions and concerns below:**
>
> > The experimental evaluation could include more diverse application domains beyond image/tabular classification.
>
> We chose to use image (high-dimensional, unstructured data) and tabular (structured) data as contrasting modalities that capture a range of real-world challenges. These experiments illustrate the effectiveness and generality of our approach. Extending the evaluation to additional modalities (e.g., text) could offer further evidence of generality, but would require additional adjustments to the problem formulation and is therefore beyond this paper’s scope. We welcome future research examining these extensions, and we also provide a discussion on regression below.
>
> > The computational overhead of the ZKP protocol may limit practical applicability in some settings.
>
> Although ZKPs add overhead for larger models, recent efficiency advances are notable. [Weng et al. 2021] achieves ZKP inference on ResNet-101 (42.5M parameters) in about 9 minutes, while [Sun et al. 2024] processes LLaMa-2-13B (13B parameters) in roughly 16.5 minutes—a 167× speedup. Moreover, startups like Polyhedra (see https://github.com/worldcoin/awesome-zkml for more examples) are rapidly advancing these techniques. As Confidential Guardian treats ZKP inference as a subroutine (line 7, Alg 1), its overhead will naturally decrease as the technology matures.
>
> > The discussion of how to set appropriate thresholds for detection in practice could be expanded.
>
> We agree that appropriately determining the detection threshold $\alpha$ is crucial in practical deployments. Appendix D.3 already provides an extensive discussion on this question, including a step-by-step guideline based on the data distribution, available domain knowledge, and existing regulatory standards. In short, we find that choosing $\alpha$ is a balance between practical constraints, domain-specific considerations, and regulatory mandates.
>
> > How would Confidential Guardian perform against adaptive adversaries?
>
> We agree that adaptive attacks are key to understanding the limits of our defense, which is why we added additional experiments in Table 2 in the appendix. As shown there, an informed attacker could try two strategies to improve Mirage’s evasiveness:
> - High $\varepsilon$: Skews the output distribution more strongly toward the correct class, making high-confidence predictions less anomalous and harder to detect via calibration metrics. However, this increases overlap with legitimate data, making selective manipulation more difficult.
> - $\varepsilon = 0$: Yields an unbiased uniform distribution, reducing confidence and accuracy jointly. While this degrades calibration-based detection, such an attack is easily identifiable with more traditional fairness evaluation metrics such as equalized odds.
>
> > How might the approach be extended to regression, generative models, or reinforcement learning?
>
> We conducted a regression experiment following the reviewer’s suggestion: https://ibb.co/kVQKYHMh. We simulate a noisy regression task and train a shallow neural network to predict a Gaussian output distribution (left plot). To adapt Mirage, we designate $[-3,-2]$ as an artificial uncertainty region in which we inflate predictive variance while keeping the mean consistent. This is done by regularizing the output variance $\sigma^{2}(x)$ towards a desired variance level $\sigma^{2}_t$:
>
> $$\mathcal{L}_{\text{penalty}}(x) = (\log \sigma^2(x)-\log\sigma^{2}_t)^2$$
>
> The right plot shows the resulting model with the uncertainty region. We will document this in more detail in a revised draft. Having validated Mirage in both classification and regression, we believe extensions to generative modeling are promising, though beyond this paper’s scope.
>
> >What are the practical considerations [...] particularly regarding computational overhead and integration with existing model serving infrastructure?
>
> Confidential Guardian is a method for detecting abstention abuse on a model during an offline phase conducted solely between the service provider and auditor. To extend its guarantees to the deployment phase, one could apply ZKPs of inference using the model committed during the audit. The specifics of such an infrastructure are an orthogonal problem which we place beyond the scope of our work, but approaches in [1] and [2] – which provide strong bounds on model properties by ZKP verifying only a small probabilistic sample of inferences – show promising scalability.
>
> [1] Franzese et al. “OATH: Efficient and Flexible Zero-Knowledge Proofs of End-to-End ML Fairness.” arXiv preprint arXiv:2410.02777 (2024).
>
> [2] Kang et al. “Scaling up Trustless DNN Inference with Zero-Knowledge Proofs.” arXiv preprint arXiv:2210.08674 (2022).
>
> **We hope that our rebuttal has addressed the reviewer’s questions/concerns and are happy to further engage with them during the discussion period.**

---

### Official Review · Reviewer_fNK9 · 2025-03-17

**Overall Recommendation:** 4

**Summary:**

This paper puts forth the possibility of discrimination by denial or delay of services using model uncertainty or confidence. The paper proposes an attack dubbed Mirage which systematically reduces confidence in a predefined uncertainty region while maintaining accuracy.  Then the paper proposes a reference dataset based way with ZKPs for detecting the same.

**Claims And Evidence:**

Yes

**Essential References Not Discussed:**

NA

**Experimental Designs Or Analyses:**

Yes

**Methods And Evaluation Criteria:**

Yes

**Other Comments Or Suggestions:**

NA

**Other Strengths And Weaknesses:**

Strengths:

1. The paper looks at a kind of discrimination by denial or delay in service which has not been looked at a lot. I find this focus refreshing.
2. The paper proposes an attack which systematically leads to reduced confidence for a target region while being accurate. I found this attack very interesting.
3. A detection mechanism using ECE and ZKP was introduced. While both these are pre-existing, I found the application to be very novel.

Weaknesses:

There are concerns over choice of reference dataset but the authors discuss this in the paper.

**Questions For Authors:**

1. The paper looks at abstention using confidence. What are some other ways of denial/delay/abstention?
2. Suppose the malicious service providers are aware of your paper and hence do not abstent with Mirage. What are some other ways of attacking with uncertainty?

**Relation To Broader Scientific Literature:**

The paper works at the intersection of security and abstention in ML. Additionally the detector works at the intersection of calibration and ZKPs. The attack is the most novel part of the paper; while other tools exist, the application is novel.

**Theoretical Claims:**

Yes

---

> ### Author Rebuttal · Authors · 2025-03-31
>
> **We thank the reviewer for their strongly positive assessment of our paper and discuss their questions below:**
>
> > What are some other ways of denial/delay/abstention?
>
> Indeed, our work considers abstention using model confidence, i.e. thresholding the maximum softmax score (as we describe in the *Abstain Option* paragraph in Section 3, see Equation (2)). While this is the predominant default abstention option due to its ease of use and broad applicability, other ways of uncertainty quantification / abstention exist. We highlight some representative instances in the *Abstention mechanisms in ML* paragraph in Section 2. After having established the foundations of artificial uncertainty induction in this paper, future work should consider the impacts of Mirage and the effectiveness of Confidential Guardian as part of other contemporary uncertainty quantification approaches.
>
> >What are some other ways of attacking with uncertainty?
>
> As the reviewer correctly points out, there are multiple ways that an uncertainty attack could be realized. While we envision many such alternate approaches, we think the most interesting alternate formulation relies on modifying the *training data* instead of the loss function. In our formulation, Mirage is designed to reduce confidence as part of the loss computation which allows for precise control of a targeted confidence level by specifying a desired $\varepsilon$. Alternatively, it would be possible to introduce uncertainty by flipping labels, injecting noise into the features, or deliberately undersampling certain parts of the dataset, and then training a predictive model with a standard cross-entropy loss on this modified dataset. This approach would work similarly to data poisoning but with the distinct goal of reducing confidence, not (just) predictive accuracy. While such an approach would likely be effective at decreasing model confidence (and other derived uncertainty metrics), we are confident that such an approach would be harder to tune to a particular targeted confidence level. Absent any loss modifications, the effectiveness of such an attack would depend on the degree of confidence the model already has on neighboring data points which might differ widely on a per-batch basis, leading to instability during training. Moreover, any non-traditional loss function adaptations would likely require a change in the amount of noise or data under-sampling which makes such an approach less flexible to use. Finally, it is unclear if such an approach would allow us to only decrease model confidence or whether it would lead to a decrease in both model confidence *and* model accuracy at the same time. Similarly to the point that the reviewer raised above, we think that this could be an exciting avenue for future work!
>
> **We hope that our rebuttal has addressed the reviewer’s questions and concerns and are happy to further engage with them during the discussion period.**

---

> > ### Comment · Reviewer_fNK9 · 2025-04-04
> >
> > Thank you for your response. I believe my rating adequately reflects my support for the paper.

---

> > > ### Author Response · Authors · 2025-04-06
> > >
> > > We thank the reviewer for their engagement in the rebuttal process and are happy to hear about their continued strong support for our paper!

---

### Decision · Program_Chairs · 2025-05-01

**Decision:**

Accept (poster)

**Comment:**

This paper addresses the potential adversarial vulnerability of the confidence of model predictions. The authors proposed both an attack and a defense for this setup. The attack, "Mirage", demonstrates that that manipulating the prediction confidence without degrading the predictive performance is practically feasible. The defense, "Confidential Guardian", employs zero-knowledge proofs to measure the misalignment between the model's accuracy and confidence.

Reviewers agreed that the paper addresses a novel and practically relevant threat in machine learning applications. The effectivenesses of the proposed attack and defense are empirically verified with solid experiments. One major limitation is that the proposed defense method suffers from the high computational costs of zero-knowledge proof, which restricted its application to large-scale setups. However, overall this is a solid paper bringing novel insights.